# Microring resonator-assisted Fourier transform spectrometer with enhanced resolution and large bandwidth in single chip solution

S.N. Zheng[1,2], J. Zou[1,3], H. Cai[2], J.F. Song[4], L.K. Chin[1], P.Y. Liu[1], Z.P. Lin [1], D.L. Kwong[2] & A.Q. Liu[1]

Single chip integrated spectrometers are critical to bring chemical and biological sensing, spectroscopy, and spectral imaging into robust, compact and cost-effective devices. Existing on-chip spectrometer approaches fail to realize both high resolution and broad band. Here we demonstrate a microring resonator-assisted Fourier-transform (RAFT) spectrometer, which is realized using a tunable Mach-Zehnder interferometer (MZI) cascaded with a tunable microring resonator (MRR) to enhance the resolution, integrated with a photodetector onto a single chip. The MRR boosts the resolution to 0.47 nm, far beyond the Rayleigh criterion of the tunable MZI-based Fourier-transform spectrometer. A single channel achieves large bandwidth of ~ 90 nm with low power consumption (35 mW for MRR and 1.8 W for MZI) at the expense of degraded signal-to-noise ratio due to time-multiplexing. Integrating a RAFT element array is envisaged to dramatically extend the bandwidth for spectral analytical applications such as chemical and biological sensing, spectroscopy, image spectrometry, etc.

[1] School of Electrical and Electronic Engineering, Nanyang Technological University, Singapore 639798, Singapore. [2] Institute of Microelectronics, A*STAR (Agency for Science, Technology and Research), Singapore 138634, Singapore. [3] College of Science, Zhejiang University of Technology, Hangzhou 310023, China. [4] College of Electronic Science and Engineering, Jilin University, Changchun 130012, China. Correspondence and requests for materials should be addressed to H.C. (email: caih@ime.a-star.edu.sg) or to L.K.C. (email: chin0062@e.ntu.edu.sg) or to A.Q.L. (email: eaqliu@ntu.edu.sg)

Optical spectrometer is a critical instrument for spectrum analysis in various applications such as chemical and biological analysis, environment monitoring, remote sensing in satellites, hyperspectral imaging, etc. Conventional spectrometers are based on free space optical engineering technology, which are usually bulky and expensive benchtop instruments. Silicon fabrication and integration technology is a powerful platform to realize chip-scale spectrometer with high compactness, high compatibility, and low cost[1–6]. Most on-chip spectrometers are based on dispersive elements such as arrayed waveguide grating (AWG)[7–10] and planar concave grating[7,10–12], which are quite similar to the conventional grating-based counterparts. Others exploit the characteristics of photonic devices to disperse light such as photonic crystal[13,14] and random photonic structures[15], and there are also other approaches such as stationary ring resonators array[16] and speckle pattern reconstruction by spiral waveguides[17]. Although these on-chip spectrometers can achieve a relatively high resolution, their main drawback is scalability for large bandwidth due to large number of detection channels. For instance, AWG based on-chip spectrometer[8] has demonstrated a resolution of 0.2 nm ($\delta\lambda$), but it needs 50 ($N$) detection channels to obtain a bandwidth of 10 nm ($\lambda_{BW} = \delta\lambda \cdot N$). Such a huge number of detection channels not only causes the complexity of the device with $N$ photodetectors (PD), but also greatly degrades the signal-to-noise ratio (SNR).

Fourier-transform (FT) spectrometers can overcome such limitations in dispersive optical spectrometers to achieve high resolution and high SNR. FT spectrometer on silicon platform has been demonstrated using microelectromechanical technology[18,19] with comparable performance to the conventional and bulky counterparts. However, it still requires moving parts and cannot be integrated with on-chip light sources and PD, which reduces its robustness. Besides, the resolution is relatively low because of the limited traveling range of the actuator. Other on-chip FT spectrometers include stationary-wave integrated FT spectrometers (SWIFTS)[20,21] and spatial heterodyne spectrometers (SHS)[22–27], which are based on spatial interferograms. In SWIFTS, only a single stationary Mach–Zehnder interferometer (MZI) is used. A resolution of 4 nm with a bandwidth of 96 nm is demonstrated[21]. On the other hand, SHS utilizes stationary MZI array to retrieve the input spectrum from a set of under-sampled discrete spatial interferogram. Although a relatively high resolution can be achieved (~0.045 nm), the need of many MZIs (e.g., 32) increases the device size and complexity[22]. Moreover, large number of detection channels are required for both methods, resulting in a low SNR. FT spectrometers can also adopt thermo-optic (TO) effect to obtain temporal interferogram[28,29] with a demonstrated resolution in several nanometers. However, high resolution is hindered by the limited optical path length and refractive index modification in a silicon chip. As a result, it remains challenging to develop an optical spectrometer that is integrated with PD onto a single chip, achieving high resolution, large bandwidth, and high SNR.

In this paper, we demonstrate a microring resonator-assisted FT (RAFT) spectrometer, which is realized using a thermally tunable photonic MZI, cascaded with a tunable microring resonator (MRR) to enhance the resolution and integrated with a PD onto a single chip. The final resolution depends on the tuning resolution of the resonance wavelength of the MRR, which is in subnanometer level due to the ultra-narrow linewidth of the resonance peak. Hence, the resolution is dramatically boosted by the MRR far beyond the classic Rayleigh criterion of the FT spectrometer without resorting to large optical path difference (OPD) of the MZI. Compared to existing FT approaches, the RAFT spectrometer requires only a single channel to achieve high resolution and large bandwidth, allowing high SNR. We demonstrate 0.47 nm resolution in ~90 nm spectral range with low power consumption (35 mW for MRR and 1.8 W for MZI). The single-chip integrated RAFT spectrometer shows simple design and easy package capability to enable compact and robust spectrometers for various spectral analytical applications.

## Results

**Resolution enhancement with a microring resonator.** Figure 1a shows schematic of the proposed RAFT spectrometer consisting of a thermally tunable MRR and MZI operating at the fundamental quasi-transverse electric (quasi-TE) mode. A broadband light is firstly butt-coupled into the input waveguide of the MRR. Only the wavelengths satisfying the resonance condition of the MRR will be transmitted to the drop port of the MRR leaving a series of dips in its throughput port[30], as illustrated in Fig. 1b. Here, for simplicity, we choose the smallest resonance wavelength of the MRR in the detected wavelength range to make the following discussion and assume $\lambda_0$ as the initial resonance wavelength. When heater 1 is activated by an external voltage, the resonance position of the MRR will shift to $\lambda_r$, inducing a relative wavelength shift as $\Delta\lambda = \lambda_r - \lambda_0$. Figure 1c shows schematic of the resonance wavelength shift of the MRR when different heating power is applied to heater 1. Here, three tuning states are displayed. The shift $\Delta\lambda_n$ is proportional to the heating power applied to heater 1. Then the following tunable MZI will retrieve each filtered spectrum from the MRR at each tuning state by analyzing the output time-domain interferogram detected by the integrated PD. By combining all the retrieved spectra, the original input spectrum is obtained (Fig. 1d). The tunable MZI is designed to be symmetric with length of 2.46 cm for each arm to achieve a resolution $R$ of ~20 nm with moderate power consumption. Free spectral range (FSR) of the MRR is designed to be larger than the resolution value $R$ of the tunable MZI. Resonance wavelength of the MRR can be tuned by a value as small as the linewidth, i.e., the FWHM which is 0.15819 nm in our experiment. Thus, the resolution can be improved from $R$ to FWHM. Therefore, the final resolution $\delta\lambda$ of the FT spectrometer assisted by an MRR can be dramatically enhanced.

**Thermal tuning.** Both MRR and MZI are thermally tunable by exploiting TO effect. The simulation results on heat transfer in silicon-on-insulator (SOI) waveguide with TiN heater can be found in Supplementary Note 1. Isolation trenches are exploited to improve heating efficiency (Fig. 2a–c). With heating power $P$ in heater above, the static temperature of the waveguide $T$ can be written as $T_0 + k_T P$ according to the Green's functions[31], where $T_0$ is the initial temperature and $k_T$ is heating efficiency depending on device materials, structures and dimensions. The static temperature $T$ is linearly proportional to the heating power $P$ and the heating efficiency with isolation trenches is 1.6 times that without isolation trenches according to simulation results (Supplementary Fig. 2a). Therefore, with isolation trenches adoption, the heating efficiency can be effectively improved. The experimental results of influences of thermal isolation trenches on heating efficiency are presented in Supplementary Note 3. The heating efficiency can be improved to maximal 12 times according to experimental results.

In the following analysis, due to the waveguide of MRR and MZI working at the fundamental mode, we will only consider the TO tuning induced effect on effective index $n_{eff}$ of the fundamental quasi-TE mode. In practice, some effects should be considered. Firstly, the effective index $n_{eff}$ of silicon (Si) waveguide has a strong wavelength dispersion. Secondly, the TO coefficient (TOC) has a nonlinear behavior during thermal tuning. The thermal expansion due to temperature excursion also

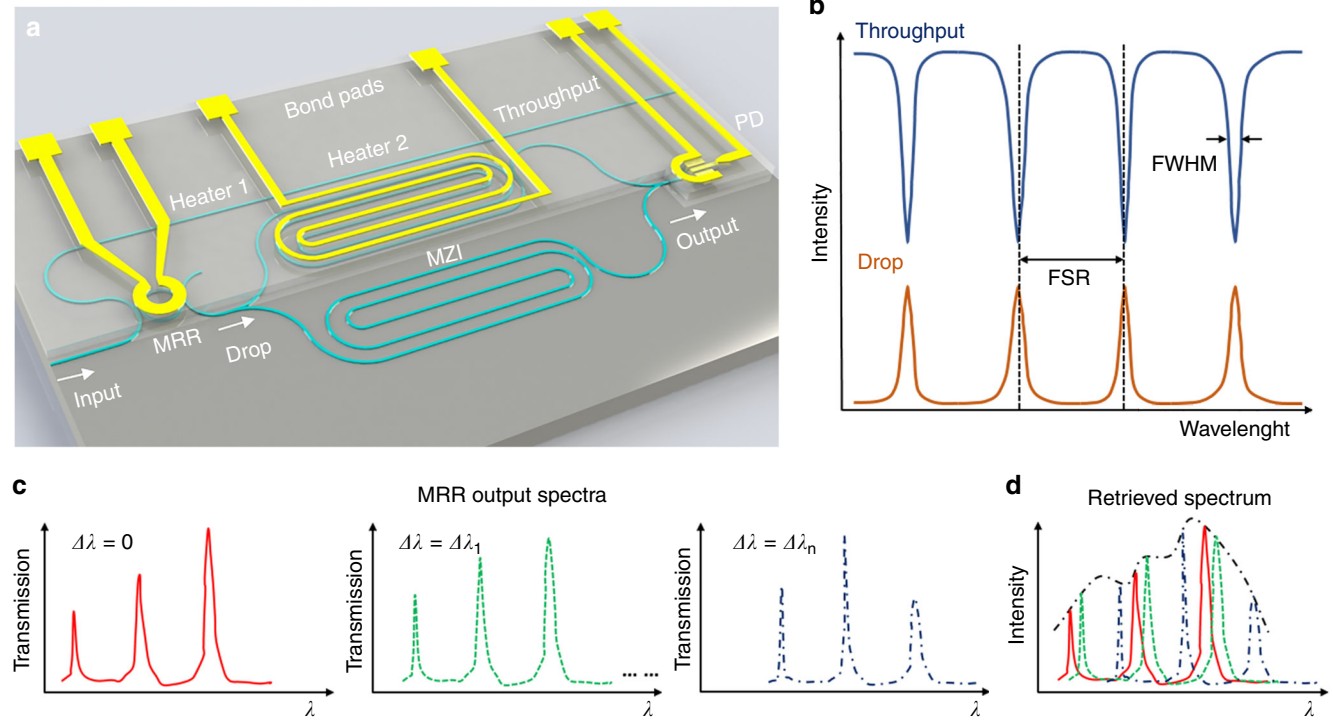

**Fig. 1** Microring resonator-assisted Fourier-transform spectrometer. **a** Schematic of the RAFT spectrometer consisting of an MRR and an MZI both with heaters on top integrated with a PD. **b** Schematic transmission spectra of the MRR. Free spectral range (FSR) is the separation between adjacent dips (peaks) and full width half maximum (FWHM) denotes the linewidth of the dips (peaks). **c** Schematic of output filtered spectra from the drop port of the MRR at three tuning states ($0 < \Delta\lambda_1 < \ldots\ldots < \Delta\lambda_n < FSR$), which are denoted by lines with different styles (colors). **d** Schematic retrieved spectra at different tuning states (denoted by lines with different styles (colors)) by the tunable MZI. The original input spectrum (denoted by black dotted line) can be retrieved by combining all the retrieved spectra

induces waveguide length change $\Delta L$. Besides, the fabrication variance will induce effective index difference ($\delta n$) and imbalance ($\delta L$) between two arms[29]. The parameter values of waveguide dispersion, TO effect and thermal expansion are presented in Supplementary Table 1. Hence, the effective index change $\Delta n_{\text{eff}}$ should be modified to

$$\Delta n_{\text{eff}} = \Delta n_{\text{eff}}(v, \Delta T) - \delta n(v), \quad (1)$$

where $\Delta T = T - T_0$ with $T_0 = 300\,\text{K}$. The total arm length difference is expressed as

$$\Delta L = \Delta L(\Delta T) - \delta L. \quad (2)$$

The expressions of $\Delta n_{\text{eff}}(v, \Delta T)$, $\delta n(v)$, and $\Delta L(\Delta T)$ are presented in Supplementary Note 2. According to the above analysis, the effective index change is proportional to temperature excursion $\Delta T$ (Fig. 2d).

The resonance wavelength of the MRR is expressed as

$$\lambda_r = \lambda_{r0} + \frac{\lambda_{r0}}{n_g} \cdot \Delta n_{\text{eff}}, \quad (3)$$

where $\lambda_{r0}$ is the initial resonance wavelength and $n_g$ is the group index. The resonance wavelength is proportional to $\Delta n_{\text{eff}}$, thus proportional to temperature excursion $\Delta T$.

For the tunable MZI, the OPD between two arms varies with the heating power $P$ applied to heater 2 residing above the upper arm as shown in Fig. 1a, which will result in a different output intensity for a different $P$. With a monochromatic source $I_i(v_0)$ ($v_0 = c/\lambda_0$) as the input of RAFT spectrometer whereby $c$ is the speed of electromagnetic wave in vacuum, the output power $I_o(\delta)$ can be expressed as $B(v_0)I_i(v_0)(1 + \cos(2\pi v_0\delta/c))$[32], where $\delta$ is OPD. The coefficient $B(\sigma_0)$ can be expressed as $0.5H(v_0)G(v_0)T(v_0)$. The factor $T(v)$ is the wavelength-dependent transmission

factor of MRR. For a given MRR, $T(v)$ is a constant for a certain wavelength. $H(v_0)$ and $G(v_0)$ are wavelength-dependent correction factors for imperfect beam splitters and optical losses, respectively. The output power consists of a constant portion $B(v_0)I_i(v_0)$ and a modulated portion $B(v_0)I_i(v_0)\cos(2\pi v_0\delta/c)$. The modulated portion constitutes the interferogram where the intensity changes with OPD. Thus, for a broadband input source, taking only the modulated portion, the output power intensity is expressed as

$$I_o(\tau) = \int_{-\infty}^{+\infty} B(v)I_i(v)\cos(2\pi v\tau)dv, \quad (4)$$

where $\tau = \delta/c$. When FT is performed to Eq. (4), the input intensity can be retrieved as

$$I_i(v) = \frac{2}{B(v)}\int_0^{+\infty} I_o(\tau)\cos(2\pi v\tau)d\tau. \quad (5)$$

Taking account of waveguide dispersion, temperature dependent TOC and thermal expansion in Eq. (4), we obtain

$$I_o(\Gamma) = \frac{1}{1 + \xi_1}\int_{-\infty}^{+\infty} B(u)[I_i(u)\cos(\varphi(u))]\cos(2\pi u\Gamma)du, \quad (6)$$

where the definitions and values of the parameters $u$, $\xi_1$, and $\Gamma$ are presented in the Supplementary Note 2. Thus, the modified input

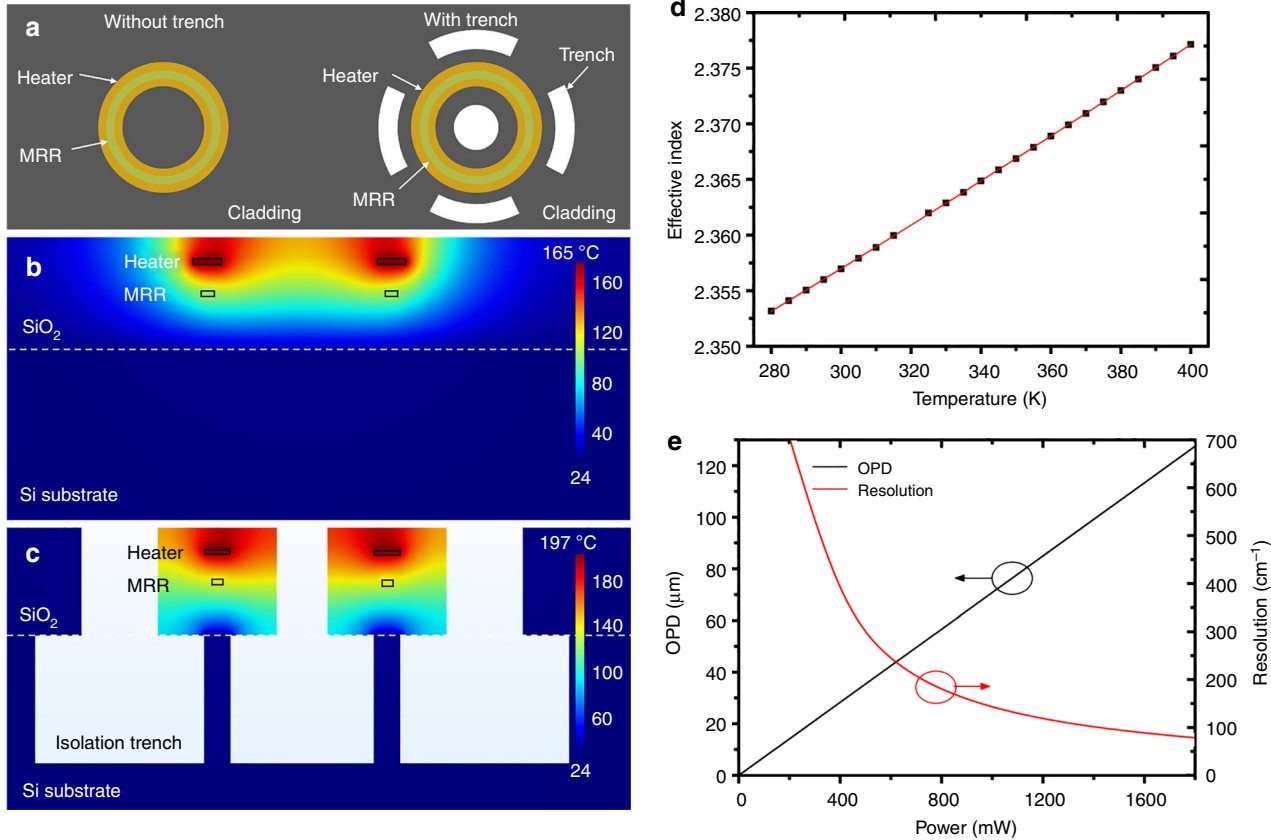

**Fig. 2** Silicon-on-insulator (SOI) waveguide thermal tuning. **a** Schematic top views of a tunable MRR without and with isolation trenches. Simulated static temperature distribution in the cross-section of a tunable MRR with constant heating power $P$ on the heater above **b** without isolation trenches and **c** with isolation trenches. **d** Relation between effective index of the quasi-TE mode and temperature. **e** Relation between OPD, resolution of the tunable MZI, and heating power on MZI heater 2

spectrum in Eq. (5) can be expressed as

$$I_i(u) = \frac{2(1+\xi_1)}{B(u)} \int\limits_0^{+\infty} I_o(\Gamma) \cos(2\pi u\Gamma) d\Gamma. \qquad (7)$$

Finally, the original input spectrum is reconstructed by transforming $u$ to $v$,

$$I_i(u) \xrightarrow{v = \frac{u - v_0}{1 + \xi_1} + v_0} I_i(v). \qquad (8)$$

$B(\sigma)$ can be obtained through experimental power calibration. The resolution of the tunable MZI $R$ is given by $1/\Delta$ where $\Delta$ is the maximum OPD. OPD equals to $\Delta n_{eff}(L + \Delta L)$. Therefore, OPD is proportional to the heating power applied to heater 2, while the resolution value decreases with increasing heating power (Fig. 2e). Simulations at different conditions (Supplementary Table 2) show the resolution of the tunable MZI can be improved either by increasing the arm length and/or increasing the heating efficiency (Supplementary Fig. 10).

The frequency information of the input spectrum of the tunable MZI can be extracted by performing fast FT (FFT) to the output interferogram (intensity changes with the applied electric power on MZI heater 2). Since the MRR prefilters the input spectrum to sparsely spaced wavelength components, the tunable MZI can differentiate the wavelength components if its resolution value is smaller than the FSR. The MRR resonance wavelength can be shifted by applying electric power on the MRR heater 1 with a tuning value as small as the FWHM. Thus, the final resolution of the RAFT spectrometer $\delta\lambda$ is dramatically enhanced compared to the designed resolution of the tunable MZI $R$, which

eases the requirement on maximal OPD for the tunable MZI. The bandwidth can be further improved by paralleling the RAFT element array with each designed specifically for a certain spectral range.

**MRR characterization**. Figure 3a shows the false-colored optical micrography of the fabricated RAFT spectrometer. Figure 3b shows the SEM image of the MRR without $SiO_2$ upper cladding, while Fig. 3c shows its final image with isolation trenches and TiN heater. Isolation trenches near MRR are exploited to improve heating efficiency and reduce thermal crosstalk between the MRR heater and MZI heater. Figure 3d shows the optical micrography of a waveguide-coupled Ge-on-SOI PD.

The experiment setup for MRR characterization is illustrated in Supplementary Fig. 17a. Based on the transmission spectrum from throughput port of the MRR (Fig. 4a), the resonance wavelengths are $\lambda_{on1} = 1528.256$ nm, $\lambda_{on2} = 1555.776$ nm, and $\lambda_{on3} = 1584.296$ nm. The measured FSR is approximately 28 nm and the linewidth (FWHM) at 1528.256 nm is ~0.15819 nm with a quality factor ($Q$) of approximately 9661. The tuning power consumption is 1.23 mW nm$^{-1}$ with a maximum estimated temperature change of 188.8 K and thus the MRR heater efficiency is around $5.5 \times 10^3$ K W$^{-1}$. Subsequently, the transmission spectra within one FSR are monitored as shown in Fig. 4b while the applied voltage on heater 1 is increased from 0 to 4.4 V. The experimental data of resonance wavelength and the heating power on heater 1 can be well fitted with a linear equation as shown in Fig. 4c. Hence, we assume a linear relation between $\lambda_r$ and heating power $P$ on heater 1.

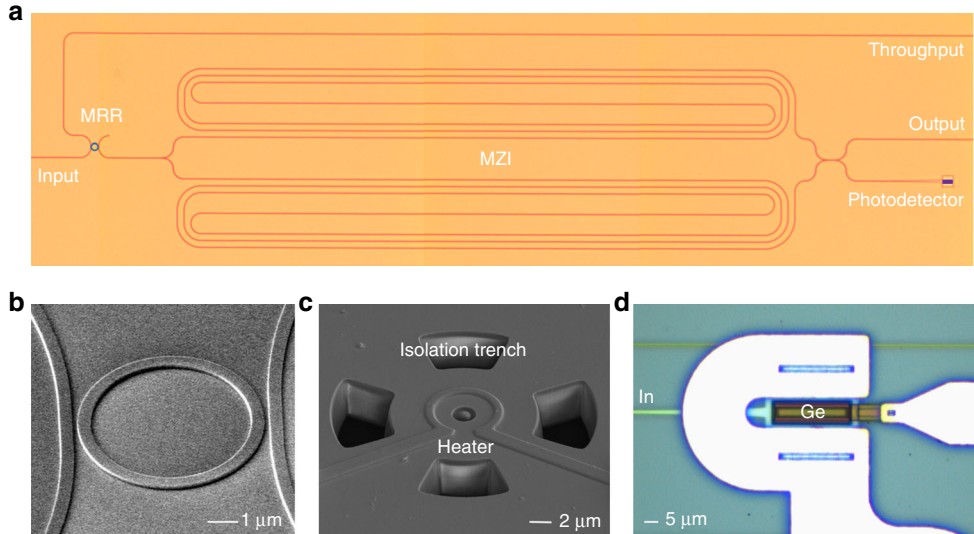

**Fig. 3** Fabricated RAFT spectrometer. **a** False-colored optical micrography of a RAFT spectrometer after Ge epitaxy growth for PD. **b** SEM image of an MRR without SiO₂ upper cladding. **c** SEM image of a tunable MRR with isolation trenches and TiN heater. **d** Optical micrography of a waveguide-coupled Ge-on-SOI photodetector

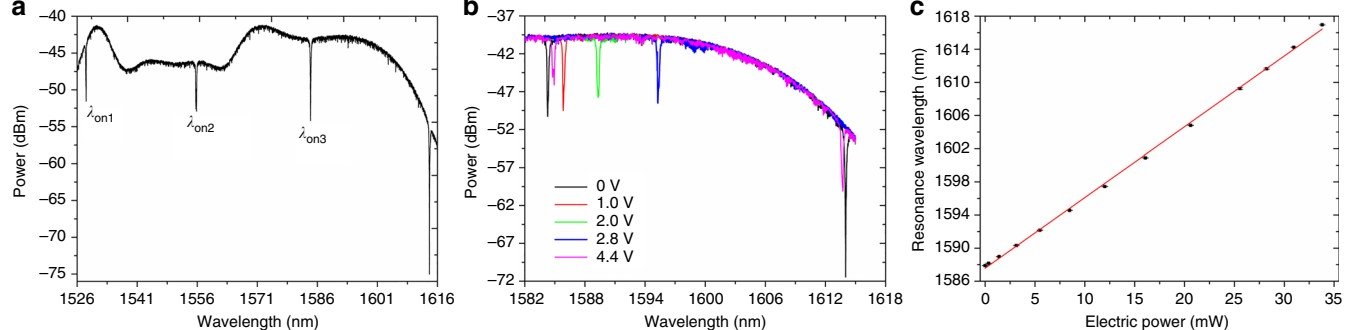

**Fig. 4** MRR characterization. **a** Transmission spectrum from the throughput port of MRR. The three resonance wavelengths are denoted as $\lambda_{on1}$, $\lambda_{on2}$, and $\lambda_{on3}$, respectively. **b** Transmission spectra within one FSR at different applied voltages on heater 1. **c** Relation between resonance wavelength and heating power on heater 1. The error bars denote S.D.

**Single wavelength characterization**. The experiment setup for RAFT spectrometer characterization is shown in Supplementary Fig. 17b. A tunable laser source (TLS-1: Santec TSL-510) is used for single wavelength characterization. Firstly, heating power is applied to heater 1 above the MRR to induce a resonance wavelength shift of $\Delta\lambda = 3.38$ nm compared to the initial static state, thus the MRR has one resonance wavelength at 1584.620 nm. Then, TLS-1 with wavelength set at 1584.620 nm and power of 6 mW is fed into the RAFT spectrometer. Figure 5a shows the detected interferogram from MZI output port. The intensity changes with the applied power on heater 2. More than 80 periods are observed with a maximum OPD of approximately 128.354 μm, which corresponds to a theoretical resolution of 77.91 cm⁻¹ (19.32 nm at 1584.62 nm). The power consumption of MZI is approximately 11.185 mW π⁻¹ with heater heating efficiency $k_T = 13.8$ K W⁻¹. Hence, the estimated maximum temperature excursion is $\Delta T = 24.9$ K. The calibration of absolute optical frequency $v$, $\gamma_2$, and $\xi_1$ and calculations of $k_T$ and $\Delta T$ are presented in Supplementary Note 2. The simulation results of relation between OPD, the resolution of the tunable MZI-based FT spectrometer and heating power on heater 2 on different conditions are presented in Supplementary Note 2. OPD is proportional to the heating power while the resolution value decreases with increasing heating power. The OPD and resolution

are improved with increasing heating efficiency $k_T$ shown in Supplementary Fig. 10a, and/or with increasing arm length $L$ shown in Supplementary Fig. 10b.

We test the resolution of the RAFT spectrometer using TLS-1. Here, we employ three resonance peaks ($\lambda_{on1} < \lambda_{on2} < \lambda_{on3}$) of the MRR to filter the input source. For simplicity, we define a detuning wavelength $d\lambda$ as $\lambda_{off}$–$\lambda_{on}$ indicating the difference between off-resonance wavelength $\lambda_{off}$ and on-resonance wavelength $\lambda_{on}$. We compare the retrieved power intensity of $\lambda_{on}$ and $\lambda_{off}$ after FFT. The power of TLS-1 is set at 8 mW. The MRR is tuned to $\Delta\lambda = 3.38$ nm. Figure 5b shows the retrieved spectra when $\lambda_{on3} = 1584.620$ nm and the value of $d\lambda$ is set to be 0.3 and 0.47 nm, respectively. One can see that when $d\lambda = 0.3$ nm, the retrieved power ratio between the on-resonant wavelength $\lambda_{on}$ and the detuned off-resonant wavelength $\lambda_{off}$ equals to 6.85 dB, while when $d\lambda = 0.47$ nm, the retrieved power ratio is increased to 10.10 dB. Similarly, the retrieved power ratios at $d\lambda = 0.47$ nm for the other two resonance wavelengths (i.e., $\lambda_{on1} = 1528.488$ nm and $\lambda_{on2} = 1556.020$ nm) are 14.77 and 16.46 dB, respectively. To effectively filter out the detuned off-resonant components into the drop port, we define the retrieved power ratio should be larger than 10 dB, i.e., the minimum MRR tuning value is 0.47 nm. Hence, the resolution of the RAFT spectrometer is defined as 0.47 nm. Figure 5c shows the retrieved spectra with TLS-1 set at

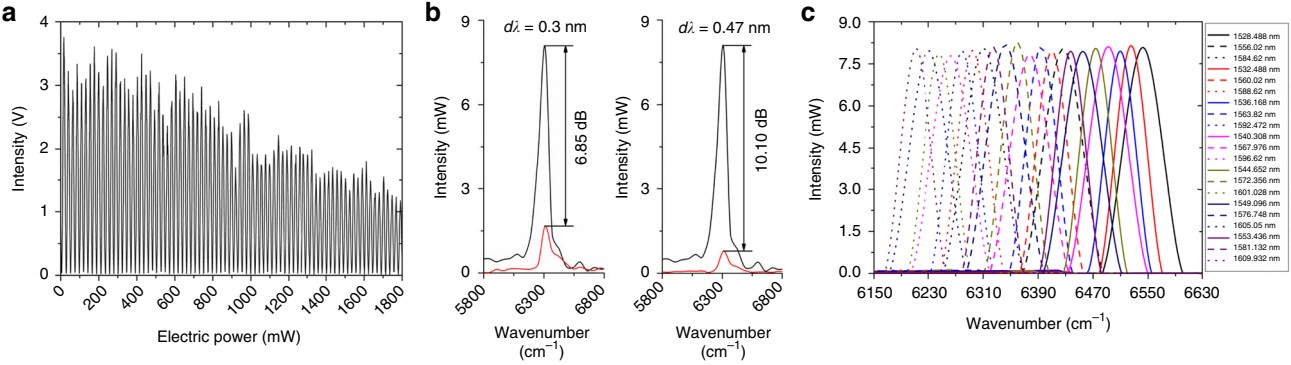

**Fig. 5** Single wavelength characterization. **a** Detected output interferogram with TLS-1 input (set at 1584.620 nm) with 6 mW input power when resonance wavelength shift $\Delta\lambda = 3.38$ nm. **b** Retrieved spectra with TLS-1 input at 8 mW input power when $\Delta\lambda = 3.38$ nm. The on-resonance wavelength $\lambda_{on3} = 1584.620$ nm and the value of $d\lambda$ is set to be 0.3 and 0.47 nm, respectively. On-resonance wavelengths are denoted in black and off-resonance wavelengths in red. **c** Retrieved spectra with TLS-1 input at on-resonance wavelengths when $\Delta\lambda = 3.38$, 7.38, 11.38, 15.38, 19.38, 23.38, and 27.38 nm, respectively

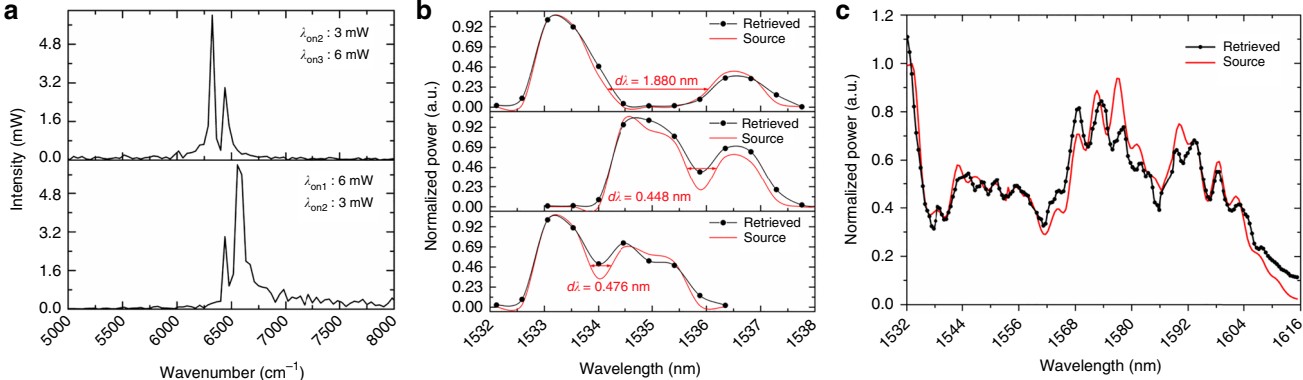

**Fig. 6** Double wavelengths and broadband spectrum characterization. **a** Retrieved spectra with TLS-1 and TLS-2 (set at adjacent on-resonance wavelengths of MRR, respectively) input simultaneously when $\Delta\lambda = 0$. **b** Normalized retrieved spectra (black) using the spectra with two broad spectral peaks input (red). **c** Normalized retrieved spectrum (black) with a broadband source (red) input. The source spectrum is generated from an optical fiber interferometer

on-resonance wavelengths of the MRR when $\Delta\lambda = 3.38$, 7.38, 11.38, 15.38, 19.38, 23.38, and 27.38 nm, respectively. It is shown that the tunable MZI can retrieve each filtered spectrum from the drop port of the MRR with single wavelength input by thermal tuning within one FSR of 28 nm.

**Double wavelength characterization**. TLS-1 and TLS-2 (ANDO AQ4321D) are combined with a 50/50 optical coupler as the input for double wavelength characterization. Figure 6a shows the retrieved spectra when TLS-1 and TLS-2 are set at adjacent on-resonant wavelengths. The tuning state is resonance wavelength shift $\Delta\lambda = 0$. The resonance wavelengths are $\lambda_{on1} = 1525.400$ nm, $\lambda_{on2} = 1552.844$ nm, and $\lambda_{on3} = 1581.240$ nm, respectively. One can see that the two adjacent on-resonant wavelength components can be easily distinguished and reconstructed by the tunable MZI. As a result, the tunable MZI can retrieve each filtered spectrum at each tuning state of the MRR.

**Broadband spectrum recovery**. To further test the resolution, we use a wavelength-division multiplexer (Sharetop WDM) to generate two broad spectral peaks as the input spectrum of the RAFT spectrometer[27,33]. The normalized retrieved spectra and the input spectra are shown in Fig. 6b. It can be seen that the minimum resolvable wavelength detuning is 0.448 nm, which is smaller than the minimum MRR tuning value of 0.47 nm. Hence, the resolution is 0.47 nm, which significantly outperforms the Rayleigh

criterion of the tunable MZI (19.32 nm). Furthermore, we performed a broadband signal measurement with minimum MRR tuning value of 0.47 nm. The transmission spectrum from an optical fiber interferometer is used as the input. For broadband signal input, all the detected input sparse spectra are retrieved using the normalization coefficient matrix $A$ (see Supplementary Fig. 9) and are then combined to produce the original input spectrum. The normalized retrieved spectrum and input broadband source are shown in Fig. 6c. The retrieved spectrum agrees well with the input spectrum. The small discrepancy is due to misalignment between lensed fiber and inverse-taper waveguide coupler while heating MZI. Another reason is resonance position fluctuation due to thermal crosstalk (see Supplementary Note 3). By packaging the lensed fiber to the input waveguide, the misalignment would not present. The thermal crosstalk mainly originates from silicon substrate since the buried oxide layer (BOX) is not thick enough to effectively isolate the heat from the heater above the MZI to Si substrate (as in our experiment, the BOX is 2 μm). By employing isolation trenches around MRR and thermal compensation (see Supplementary Note 3), the stability tolerance of the resonance wavelength has been decreased from $2\delta\lambda$ (without thermal compensation) to $\delta\lambda/5$. The current value can be further decreased by reducing residue thermal crosstalk, optimized thermal compensation and/or adopting heater with low-temperature coefficient of resistance. The residue thermal crosstalk can be further mitigated through fabricating isolation trenches near MZI arms and can also be effectively reduced by

making both MRR and MZI fully suspended[34] (see Supplementary Note 3) and/or using thicker BOX. Moreover, it will be well compensated with a feedback circuit to control the applied power on MRR and MZI heater.

## Discussion

The designed RAFT spectrometer consisting of a tunable MRR and a tunable MZI enhances the resolution dramatically far beyond the Rayleigh criterion of a typical tunable MZI (42.6-fold here). Due to the employment of MRR filter, the MZI only needs a resolution (<28 nm) to resolve the resonance wavelengths with a minimum span of one FSR of the MRR, significantly easing the requirement on the maximal OPD. Since thermal isolation trenches are employed for the MRR, the power consumption is significantly reduced. The total energy consumed by MRR and MZI for $N$ scans are 1.96 and 67.2 J, respectively. The calculations are presented in Supplementary Note 3. The power consumption of MZI can be reduced to 150 mW through fabrication of isolation trenches along the waveguides of MZI arms (see Supplementary Note 3). Hence, the total energy consumed by MZI can be reduced from 67.2 to 5.6 J. Note that the Si substrate under the MRR in the tested RAFT spectrometer and the MZI arm in the testing structure is not totally removed (Supplementary Fig. 11b), the heating efficiency of both MRR and MZI can be further improved (~8.75 times) if the waveguides are fully suspended[35]. The resolution limit of the tunable MZI due to waveguide dispersion is $18 \leq R \leq 19.9$ nm in the detected wavelength range with fixed maximal OPD (with maximal heating power of 1.8 W employed in the experiment). In our proposed structure, the final resolution $\delta\lambda$ can be further improved by increasing the $Q$ value through coupler design (e.g., optimizing the gap and/or coupling length of the coupling region) and decreasing the losses in the ring waveguide and couplers via fabrication optimization. For instance, for single-pass amplitude transmission $a = 0.9986$, if $Q \geq 10,000$, i.e., $\delta\lambda \leq 0.153$ nm at 1528.256 nm, the self-coupling coefficient $r \geq 0.9835$ (see Supplementary Note 5). The parameter values for this calculation are shown in Supplementary Table 3. Hence, the gap between ring and straight waveguide is larger than 230 nm according to FDTD simulation results. Noticing that the gap dominates in determining $r$, thereby, the $Q$ value, only the fabrication tolerance of gap is considered here. Since the transmitted power from MRR will be reduced when increasing $Q$ value (see Supplementary Fig. 16b), the designed gap is 240 nm with 20 nm tolerance, i.e., ±10 nm fabrication deviation, which can be easily achieved by the current fabrication technology (±7.5 nm deviation). Although the working spectral window depends on the transmission band of various components such as waveguides, couplers, beam splitters, and PD, etc., the bandwidth can be drastically extended by designing a paralleled RAFT spectrometer array.

The MRR before the MZI will compromise the Fellgett advantage of a typical FT spectrometer, which will induce a lower SNR. The calculations are presented in Supplementary Note 4. Hence, the SNR requirement needs to be considered, since it will limit the minimum resolution value as shown in Supplementary Fig. 16c with simulation parameter values summarized in Supplementary Table 3. The multiplex gain loss is approximately 87.5% with $m = 3$ in our experiment. With decreased SNR, the level of the minimum detectable signal is increased, thus leading to reduced dynamic range. This loss can be reduced by appropriately increasing $m$, i.e., employing an MRR with larger circumference which has smaller FSR. At the same time, the resolution $R$ (equal to FSR) of the tunable MZI must be improved accordingly (see Supplementary Fig. 14). To improve resolution, a larger maximal OPD $\Delta$, i.e., more heating power and/or longer arm length are required. The increased heating power not only increases the power consumption, but also brings larger TO

non-linearity and thermal expansion effect. Moreover, the longer waveguide length will induce larger footprint and higher optical loss due to imperfect fabrication. The higher optical loss will in turn reduce the contrast of interferogram, leading to reduced SNR. It is very challenging to achieve a resolution down to sub-nm using a tunable MZI. It is suggested that the resolution of the tunable MZI, $R \geq 10$ nm, when moderate power consumption and arm length are required and resolution enhancement ($R/\delta\lambda$) is larger than 20 times. Hence, $m$ is chosen as $m \leq 9$. Time-multiplexing will also induce extra power consumption due to multiple scans of MRR and MZI. Nearly, 2% ($\delta\lambda$/FSR) of total time is spent for measuring a single resolution element and in our current experiment, the one-time scan duration is 2 s. The time-scale measurement on thermal response time of MZI and MRR is presented in Supplementary Note 4. The results show that the maximum sweeping frequency of MZI is 10 kHz. For 10 kHz sampling frequency and 2000 one-time sampling points, the one-time scan duration is reduced to 0.2 s and the total time is reduced to 0.2FSR/$\delta\lambda$ ≈11.4 s. Hence, the fast sweeping frequency of MZI will compensate the gain loss due to time-multiplexing and reduce the total energy consumption of MZI and MRR.

In conclusion, a microring RAFT spectrometer is experimentally demonstrated with a tunable MRR, a tunable MZI, and a Ge-on-SOI PD being integrated onto a single chip. The tunable MRR pre-filters the input spectrum into a sparse spectrum to match the resolution of the following cascaded tunable MZI. Due to the high-quality factor (~9661) of the MRR, the resolution of the RAFT spectrometer is dramatically boosted far beyond the Rayleigh criterion of a typical FT spectrometer by finely tuning the resonance wavelength of MRR. A high-resolution of 0.47 nm and a large bandwidth of ~90 nm is achieved. The bandwidth can be largely extended by integrating a paralleled RAFT element array. The power consumption due to thermal tuning and time-multiplexing can be drastically reduced by introducing isolation trenches and increasing the sweeping frequency of MZI. The SNR degraded by time-multiplexing can be improved by reducing optical loss and/or adopting smaller FSR. It has high potential for applications such as chemical and biological sensing, on-chip spectroscopy, and image spectrometry.

## Methods

**Fabrication**. The microring RAFT spectrometer is fabricated from an 8-inch SOI wafer using the nano-silicon photonic fabrication technology. After fabricating the Si waveguides structures, several implantation processes and Ge epitaxy are done for fabrication of the waveguide-coupled PD. Subsequently, a 1 μm-thick upper silicon dioxide (SiO$_2$) cladding layer is deposited and then a thin layer of titanium nitride (TiN) is formed to act as the resistive layer for heaters. Subsequently, an aluminum (Al) thin film is patterned for electrical connection to power the heaters and PD. At last, the isolation trenches are formed through etching SiO$_2$ cladding and Si substrate.

**Experiment setup**. The experiment setup for MRR characterization is shown in Supplementary Fig. 17a. A broadband light source (Amonics C + L Band ASE broadband light source) is used to perform MRR characterization. It is firstly coupled to a polarization beam splitter and a polarization controller to ensure that only TE-polarized light is input into the on-chip spectrometer. The device under test, i.e., the on-chip spectrometer chip with a thermo electric cooler as the substrate to control and stabilize the temperature using a temperature controller, is mounted on the XYZ stage holder for fiber-chip alignment. The output spectrum from the throughput port of the MRR is detected by an optical spectrum analyzer (OSA, Yokogawa AQ6370D). The applied voltages of heater 1 and heater 2 are controlled by a laptop via a microcontroller.

The experiment setup for spectrometer characterization (single and double wavelength characterization and broadband spectrum recovery) is shown in Supplementary Fig. 17b. TLS-1 (santec TSL-510) is adopted as the input only for single wavelength characterization. TSL-1 and TLS-2 (ANDO AQ4321D) are used simultaneously to perform double wavelength characterization. Light from TLS-1 and TLS-2 is combined with a 50/50 optical coupler as the input. The output light from MZI is detected by an off-chip photodetector (PD, Thorlabs PDA-10CS-EC) of which the intensity signal is acquired by a laptop via a microcontroller.

**Electrical measurements.** The spectrometer chip is wire-bonded to a PCB board for electric connections with an external PCB board which is subsequently connected to a microcontroller. The external PCB board is integrated with DACs and an operational power amplifier (OPA) to amplify the electrical signal for MZI heater. The resistances of MRR heater and MZI heater are ~0.547 and ~1.277 kΩ, respectively.

**Thermal compensation.** Even though the isolation trenches fabricated near MRR have effectively reduced the thermal crosstalk from MZI heater, there is still a remaining resonance wavelength shift (maximal 1 nm) due to thermal crosstalk. Hence, thermal compensation is performed to stabilize the resonance position. The thermal compensation method is presented in Supplementary Note 3.

**Broadband source calibration.** The description of the calibration process is presented in Supplementary Note 2.

## Data availability
The data that support the findings of this study are available from the corresponding authors on reasonable request.

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

## Acknowledgements
This work was supported by Ministry of Education, Singapore under Tier 3 (MOE2017-T3-1-001) and Tier 1 (RG182/16 (S)) and the Singapore National Research Foundation under the Competitive Research Program (NRF-CRP13-2014-01) and the Incentive for Research & Innovation Scheme (1102-IRIS-05-04) administered by PUB. The authors acknowledge Liu Yong Sheng for kind help and fruitful discussions.

## Author contribution
S.N.Z. and A.Q.L. jointly conceived the idea. S.N.Z., J.Z., and J.F.S. performed the numerical simulations and theoretical analysis. S.N.Z., H.C., and D.L.K. did the fabrication of the device. S.N.Z. and J.Z. did experiments of spectrometer characterization. S.N.Z., J.Z., L.K.C., P.Y.L., Z.P.L., and A.Q.L. prepared the paper. J.F.S., Z.P.L., and A.Q.L. supervised and coordinated all the work. All authors commented on the paper.

## Additional information

**Competing interests:** The authors declare no competing interests.

