## [Peer Review File · Nature Communications]

Reviewers' Comments:

Reviewer #1:

Remarks to the Author:

The authors describe a single chip integrated cavity-enhanced Fourier-transform (CEFT) spectrometer consisting of a tunable microring resonator (MRR) input filter followed by integrated thermally tunable photonic Mach-Zehnder interferometer (MZI) integrated with a photodetector. The purpose of the MRR input filter is to produce a sparse spectrum with resolution elements that are separated by FSR far enough to be resolved by the MZI, while also guaranteeing that the resolved peaks can be deconvolved to achieve the MRR resonance linewidth. The MRR filter in the cascade is used to enhance the resolution to 0.47 nm whereas the MZI allows to achieve a bandwidth of ~90 nm with heaters power consumption of 35 mW and 1.8 W for MRR and MZI heaters, respectively.

It is a clever idea, but after careful consideration, I feel that behavior of the system tends to be self-defeating when it is pushed toward higher resolution (which is its ultimate goal). This is primarily because of the interplay between FSR and FWHM in high Q ring resonators. Ultimately, I feel that this device will be outperformed by either a pure coupled ring resonator spectrometer, or a pure FTIR spectrometer. Some of specific comments below enhance this points:

1. The described device abandons many of FTIR spectrometers advantages in order to get the improved resolution. Specifically, rather than having all input signal power of a broadband input signal falling on a photodetector at once as it is commonly done in FTIR, the proposed approach significantly reduces the input signal power due to transmission through the MRR filter. In fact as the Q of the MRR filter increases to achieve high spectral resolution, the fraction of power contained in the output comb of the MRR filter decreases leading to decrease in the detector's SNR and corresponding dynamic range.

More specifically, the authors need to quantify the loss in multiplex advantage incurred by the tuning mechanism. The multiplex, or Fellgett advantage that defines the Fourier transform spectrometer can be stated as the SNR gained by the simultaneous measurement of all spectral orders over monochromator measurements. While the cavity-enhanced Fourier transform spectrometer is able to guarantee higher resolution owing to the microring resonator, the authors must factor in the time multiplexing to sweep through the spectrum of the input signal that is broad. In particular, the extremely narrow linewidth of the MRR resonances would suggest that for a given time T, the amount of time spent measuring a single resolution element of width $\Delta\lambda$ is roughly $T \cdot \Delta\lambda / \text{FSR}$. Since the device relies on the MZI resolving at least down to the MRR FSR, for the 0.47 nm linewidth and a 19.7 nm MZI resolution at 1584 nm, this means that roughly 2% of the total collection time is spent measuring this resolution element before the MRR is tuned to scan. The authors should quantify this loss in multiplex gain and explain the tradeoff made for reduced footprint, power consumption, bandwidth, etc. In particular, for the same spectral collection time of M resolution elements of width $\Delta\nu$ over a bandwidth of Ω such that $M = \Omega / \Delta\nu$, the multiplex gain over a monochromator is generally \sqrt{M} . This is because a dispersive measurement must separately measure each spectral element over the same collection time, reducing the dwell time and consequently SNR for each spectral order. In contrast, the Fourier transform spectrometer simultaneously measures all spectral components for the entirety of the same collection time, and separates the resolution elements computationally.

2. The power consumption of the device is very high. This is because each scan only covers the spectra that overlaps with the MRR resonance line. This means that to cover the full spectra of the input signal it will be necessary to perform a number of scans equal to FSR / FWHM . This number become very large as the resolution of the MRR increases (e. g., $70\text{nm} / 0.47\text{ nm} > 140$). Maybe a total energy is a better parameter, for example, with time to tune 100 usecs, the ring based spectrometer will consume about $35\text{mW} \times 140 \times 10\text{E-}4\text{sec} = 0.5\text{mJ}$ whereas the current device will consume about $1.8 \times 140 \times 10\text{E-}4 = 25\text{mJ}$. This is a great penalty. In terms of power consumption, both a pure coupled ring resonator spectrometer, and a pure FTIR spectrometer are superior to

this design.

3. The bandwidth of ring coupler seems also problematic. It will never get above 100nm or so, and the efficiency will fall at the edges. This means larger spectra will need multiple unit cells, which further compounds the power consumption challenge.

4. It should also be noted that FSR and linewidth (FWHM) are related, such that a high Q ring resonator tends to also have a small FSR. At least for a single ring, this may defeat the purpose of the overall design, because the FTIR resolution must be equal to the FSR. A multiple ring device can solve this issue, but if you have to play with multiple rings, there is not really any reason to keep the FTIR part. It would be better to just stay with a pure ring-based spectrometer.

5. Regarding the measurements, the experiments do not include a broadband signal measurement, only two fairly closely spaced narrowband laser lines are used for measurements. This is not sufficient to justify publication in high impact journal. In fact, this is not a surprise, as the authors neglect dispersion in their analysis, which would spoil an attempt to characterize a broadband input signal. The authors also neglect the benefit of dispersion in estimating the limits of resolution/temperature efficiency for a pure FTIR spectrometer. This is somewhat disappointing, making the device performance look better.

6. The calibration procedure is not clearly explained. Was the calibration performed independently for each wavelength? How will it be done for operation with broadband signals?

7. There are a number of spelling errors throughout the manuscript, and the authors should check to make sure that the units are capitalized where need be (k -> K for Kelvins, etc).

In summary, the manuscript needs a more rigorous analysis of the device performance to include dispersion, complete analysis of power consumption, detection SNR and dynamic range penalty, clearly describe the calibration procedures, and perform a broadband signal measurements.

Reviewer #2:

Remarks to the Author:

The manuscript presents an integrated device based on a silicon photonics platform, i.e., Si/SO₂ waveguides obtained in SOI substrates. The device is comprised of an input ring resonator, followed by a Mach Zehnder interferometer (MZI) and a final integrated Ge-based photodetector. Both the ring resonator and one of the arms of the MZI are subject to index of refraction control via electrical heating. The main idea of the integrated device is to use the fine resonances of the ring resonator to filter the input spectrum which is the object of analysis. The filtered input is sent to the MZI where one of the arms has its optical path changed by heating. The combined light from both MZ arms is sent to the photodetector. An inverse fast Fourier transform (IFFT) of the photodetector signal, considering time and or heating power, leads to the recovery of the filtered spectrum. By applying heat to the ring resonator, one can move the resonance peaks to a different part of the spectrum under analysis and perform again the IFFT approach again. In this fashion, the manuscript claims that one can recover the spectrum with high resolution, limited by the Q of the ring resonator, not as much as on the MZ optical path difference between the two arms. Also, it is claimed that a larger range can be achieved due to the large tunability of the ring resonator as well as by a proposed construction of a more complex integrated device where many similar structures would be fabricated side by side.

The idea of using the ring resonator to improve resolution is not new, but the combination of the three structures, i. e., ring resonator, interferometer, photodetector is indeed challenging and impressive. Nevertheless, I believe several serious issues have to be addressed before the manuscript can be considered for publication. I list these issues below:

1) By using the ring resonator as a filter, it is essential that there is no change in the transmission value for the different resonances at a given temperature as well as no variation as a function of the temperature tuning. The characterisation of the resonances and the transmission properties of the input ring resonator have to be addressed both theoretically and experimentally.

2) The authors mention the heat isolation trenches to improve the heating/thermal optical effect efficiency. Thermal crosstalk between the MZ and the ring resonator is also a major problem, because the operation of the former might affect the resonance position of the ring resonator. This is not addressed neither theoretically nor experimentally in the manuscript.

3) In the thermal modulation session, absolutely no non-linearity in the thermal optical effect (also temperature expansion) is considered. The analysis is very simplistic. In fact, given the excursion of the MZ thermal optical effect (1.8 W and temperatures up to 150 C) it is essential that non-linearities are taken care of as it has been in recent published in Nature Comm on the same subject.

4) In the single wavelength characterisation, there are several important issues to be considered.

4.1) Firstly, the acronym TLS is never defined. I assume is tunable laser source. This is indeed simple to be corrected.

4.2) There is a reference to a resonance shift that is indeed very confusing. What is this shift? Is it caused by the input light power? The entire discussion of the resonance shift which is done onwards is very difficult to follow. This has to be very, very, well clarified.

4.3) The authors mention that under a single wavelength input one observes a sinusoidal behaviour of the photodetector current, as shown in Fig. 5a. The behaviour is absolutely not sinusoidal. There is a large amplitude variation, plus, although not shown, there should be chirp which would explain the large difference between the retrieved spectrum with respect to the input. This may not appear to be a big problem when a source with only one or two wavelengths are employed. However, this would make the device unusable for a continuous reconstruction. This is the most serious negative issue of this manuscript. There should be a demonstration of a wide spectrum recovery. Based on the results for single or double wavelength, it appears that the non-linearities that I commented above, if not considered and taken care of, will make the device unusable. Therefore, a better evaluation of the non-linearity and a demonstration of a wide spectrum recovery is paramount for the demonstration of the device functionality.

Considering the comments above, unless each and every one of them are carefully considered, the manuscript should not be accepted for publication.

Reviewer #3:

Remarks to the Author:

The manuscript "A Cavity-enhanced Fourier Transform Spectrometer with High Resolution and Large Bandwidth in Single Chip Solution" proposes cascading a tunable micro-ring resonator (MRR) in front of a conventional on chip tunable Mach-Zehnder interferometer (MZI), utilized as a Fourier transform spectrometer, to increase the spectral resolution. The authors claim that their spectrometer can reach a resolution of 0.47 nm (equal to the FWHM of the MRR) while the nominal resolution of the MZI is only 20 nm.

General comments:

1- The term "Cavity-enhanced" in the title is not used in its conventional manner. The enhancement cavity in spectroscopy is generally used for increasing the light-matter interaction length and thus boosting the sensitivity of the spectrometer, while the authors used this term to state the enhancement in the spectral resolution of the spectrometer. I personally find this misleading and strongly suggest either to remove it from the title or to rephrase it.

2- Although the idea of the authors in cascading a MRR with a MZI to increase the spectral resolution seems to be novel, unfortunately they fail to demonstrate this enhancement in practice. In Fig. 4 (a) the authors show the transmission spectrum of the MRR with ~ 0.5 nm FWHM, measured with an optical spectrum analyzer, not the MZI. In Fig. 6 the measured spectrum of two cw laser sources (with a wavelength difference of ~ 30 nm) are shown and it is obvious from the figure that the spectral resolution of the spectrometer is ~ 20 nm. The authors do not demonstrate the possibility of resolving two spectral features with a separation in the order of their claimed resolution (~ 0.5 nm). In addition the process of extracting a high resolution broadband spectrum by their spectrometer is not explained and there is no comments about the time scale needed for performing such a measurement. The measurement time is specifically important since the performance of a heater-based spectrometer is compared with an AWG-based one (see [5-8]).

3- There are several grammatical and phrasing errors in the text that at some cases even make it hard to understand the general point. I strongly recommend the authors to ask a native speaker to review and refine the manuscript.

In conclusion, unfortunately I do not believe that the manuscript – on its present status – is appropriate to be published in nature communications. The main claim of the article is not shown to be achieved, and the presentation needs to be improved drastically.

Manuscript ID: NCOMMS-18-23000A

Paper title: A Cavity-enhanced Fourier Transform Spectrometer with High Resolution and Large Bandwidth in Single Chip Solution

Authors: S.N. Zheng, H. Cai, J. F. Song, L. K. Chin, P. Y. Liu, Z. P. Lin, D. L. Kwong, and A. Q. Liu

Reply to Reviewer 1

We are grateful to the Reviewer for the constructive comments and are happy to address all the comments.

Comment 1: *In fact as the Q of the MRR filter increases to achieve high spectral resolution, the fraction of power contained in the output comb of the MRR filter decreases leading to decrease in the detector's SNR and corresponding dynamic range. More specifically, the authors need to quantify the loss in multiplex advantage incurred by the tuning mechanism.*

Reply: As pointed out by the reviewer, the calculation of the multiplex gain loss is added in revised Supplementary Note 4. The discussion on how this loss influences SNR and dynamic range is added to revised manuscript as “The loss is approximately 87.5% with $m = 3$ in our experiment, which will induce decreased SNR. With decreased SNR, the level of the minimum detectable signal is increased, thus leading to reduced dynamic range.” in Line 295 Page 14.

While the cavity-enhanced Fourier transform spectrometer is able to guarantee higher resolution owing to the microring resonator, the authors must factor in the time multiplexing to sweep through the spectrum of the input signal that is broad.

Reply: As pointed out by the reviewer, time spent for measuring a single resolution element is $\delta\lambda/\text{FSR} \approx 2\%$ before the MRR is tuned to scan. The experimental measurement of thermal responses of MZI and MRR are added to revised Supplementary Note 4 and the thermal response time of MZI and MRR are shown in Supplementary Fig. S15. The analysis on influences of time-multiplexing is added to revised manuscript as “Time-multiplexing will also induce extra power consumption due to multiple scans of MRR and MZI. Nearly 2% ($\delta\lambda/\text{FSR}$) of total time is spent for measuring a single resolution element and in our current experiment, the one-time scan duration is 2 s. The time-scale measurement on thermal response time of MZI and MRR is presented in Supplementary Note 4. The results show that the maximum sweeping frequency of MZI is 10 kHz. For 10 kHz sampling frequency and 2,000 one-time sampling points, the one-time scan duration is reduced to 0.2 s and the total time is reduced to $0.2\text{FSR}/\delta\lambda \approx 11.4$ s. Hence, the fast sweeping frequency of MZI will compensate the gain loss due to time-multiplexing and reduce the total energy consumption of MZI and MRR.” in Line 309 Page 15.

Supplementary Fig. S15: Thermal response time of **a** tunable MZI and **b** tunable MRR.

The authors should quantify this loss in multiplex gain and explain the tradeoff made for reduced footprint, power consumption, bandwidth, etc.

Reply: A MRR is employed and the tunable MZI only needs to resolve the wavelengths with a separation equal to FSR of the MRR. This drastically enhances the resolution by fine tuning the MRR and eases the requirement on the maximum OPD at the same time. Moreover, due to the small size of the MRR, its power consumption of ~ 35 mW is far less than that of the MZI (i.e., 1.8W) in our experiment. Therefore, the proposed scheme effectively decreases the power consumption and footprint compared to other thermally tunable MZI-based FTIR as in [26] in the reference despite the loss in multiplex gain.

As suggested by the reviewer, the discussion on the loss is added to revised manuscript as “This loss can be reduced by appropriately increasing m , i.e., employing a MRR with larger circumference which has smaller FSR. At the same time, the resolution R (equal to FSR) of the tunable MZI must be improved accordingly (see Supplementary Fig. S14). To improve resolution, a larger maximal optical path difference Δ , i.e. more heating power and/or longer arm length are required. The increased heating power not only increases the power consumption, but also brings larger thermo-optic non-linearity and thermal expansion effect. Moreover, the longer waveguide length will induce larger footprint and higher optical loss due to imperfect fabrication. The higher optical loss will in turn reduce the contrast of interferogram, leading to reduced SNR. It is very challenging to achieve a resolution down to sub-nm using a tunable MZI. It is suggested that the resolution of the tunable MZI, $R \geq 10$ nm, when moderate power consumption and arm length are required and resolution enhancement ($R/\delta\lambda$) is larger than 20 times. Hence, m is chosen as $m \leq 9$.” in Line 297 Page 14.

Comment 2: *The power consumption of the device is very high. Maybe a total energy is a better parameter, for example, with time to tune 100 usecs, the ring based spectrometer will consume about $35\text{mW} \times 140 \times 10^{-4} \text{sec} = 0.5\text{mJ}$ whereas the current device will consume about $1.8 \times 140 \times 10^{-4} = 25\text{mJ}$. This is a great penalty. This is a great penalty. In terms of power consumption, both a pure coupled ring resonator spectrometer, and a pure FTIR spectrometer are superior to this design*

Reply: As suggested by the reviewer, the calculation for the total energy consumed by MRR and MZI is added to revised Supplementary Note 3. The one-time scan duration is 2 s and $N = 56$ in the experiment. The total energy consumed by MRR, $P_{r\text{-total}} = 1.96$ J and the total energy consumed by MZI, $P_{m\text{-total}} = 67.2$ J.

As pointed out by the reviewer, a pure coupled ring spectrometer and/or a pure FTIR is superior to this design in terms of power consumption. The proposed design achieves both high resolution and large bandwidth using a tunable ring, MZI and PD all integrated on a single chip with a compact size of 5.5×0.4 mm². It is hard for a pure coupled ring spectrometer or a pure FTIR to achieve high resolution and large bandwidth at the same time.

Considering the power consumption of MZI, we can employ isolation trenches near MZI arms in the chip fabrication to improve the heating efficiency. MZI testing structures (Supplementary Fig. S11a) are fabricated to test the influence of isolation trenches on heating efficiency. The measurement and results are added to revised Supplementary Note 3. The heating efficiency can be improved to maximum 12 times by either reducing the gap and/or increasing the trench segment length L . The maximum heating power can be reduced to $1.8/12$ W = 150 mW and the total consumed energy of MZI can be reduced to $P_{m\text{-total}}/12 = 5.6$ J. The consumed power is significantly decreased compared to other silicon-based thermally tunable MZI as in [26] in the reference. Besides, it can be further reduced (~8.75 times) when the waveguides are fully suspended referring to [31] in the reference.

The discussion on power consumption is added to revised manuscript as “The total energy consumed by MRR and MZI for N scans are 1.96 J and 67.2 J, respectively. The calculation is presented in Supplementary Note 3. The power consumption of MZI can be reduced to 150 mW through fabrication of isolation trenches along the waveguides of MZI arms (see Supplementary Note 3). Hence, the total energy consumed by MZI can be reduced from 67.2 J to 5.6 J. Note that the Si substrate under the MRR in the tested RAFT spectrometer and the MZI arm in the testing structure is not totally removed (Supplementary Fig. S11b), the heating efficiency of both MRR and MZI can be further reduced (~8.75 times) if the waveguides are fully suspended^[31].” in Line 276 Page 14.

Supplementary Fig. S11: **a.** Schematic of MZI testing structure with isolation trenches. **b.** SEM image of the cross section of the thermal isolation trench. **c.** Power consumption vs gap between the trench and waveguide.

Comment 3: *The bandwidth of ring coupler seems also problematic. It will never get above 100nm or so, and the efficiency will fall at the edges. This means larger spectra will need multiple unit cells, which further compounds the power consumption challenge.*

Reply: As pointed out by the review, the transmission of the ring coupler is wavelength dependent and it is very hard to keep the same efficiency across the 100-nm wavelength bandwidth. Multiple unit cells are required to extend the spectral range, which further increases the power consumption. Hence, we must reduce the power consumption by such as improving MZI sweeping frequency and/or improving heating efficiency with optimized isolation trenches.

Comment 4: *It should also be noted that FSR and linewidth (FWHM) are related, such that a high Q ring resonator tends to also have a small FSR. At least for a single ring, this may defeats the purpose of the overall design, because the FTIR resolution must be equal to the FSR.*

Reply: For a single ring, the value of quality factor Q is expressed as [1]

$$Q = \frac{\pi}{FSR} \cdot \frac{\lambda \sqrt{r_1 r_2} a}{1 - r_1 r_2 a} \quad (1)$$

where r_i ($i = 1,2$) is the self-coupling coefficient of ring coupler and a is the round-trip loss coefficient (no loss when $a = 1$). As indicated in Eq. 1, the Q-value is mainly determined by the coupling parameters r_1 and r_2 and the round-trip loss coefficient a , while the FSR value contributes little to Q in comparison to r_1 , r_2 and a . High Q value can be achieved by making the denominator of the second term on the right, i.e., $1 - r_1 r_2 a$, trend to zero, which can be realized by increasing r_1 and r_2 to approach 1 through coupler design (e.g. optimizing the gap and/or coupling length of the coupling region), and decreasing the losses in the ring waveguide and couplers to make a closer to 1 via fabrication optimization. The discussion is added to the revised manuscript as “In our proposed structure, the final resolution $\delta\lambda$ can be further improved by increasing the Q-value through coupler design (e.g. optimizing the gap and/or coupling length of the coupling region) and decreasing the losses in the ring waveguide and couplers via fabrication optimization.” in Line 286 Page 14.

[1] W. Bogaerts, P. De Heyn, T. Van Vaerenbergh, K. De Vos, S. Kumar Selvaraja, T. Claes, P. Dumon, P. Bienstman, D. Van Thourhout and R. Baets, *Laser & Photonics Reviews* 6, 47-73 (2012).

A multiple ring device can solve this issue, but if you have to play with multiple rings, there is not really any reason to keep the FTIR part. It would be better to just stay with a pure ring-based spectrometer.

Reply: For multiple rings-based spectrometer, there is a stringent requirement on the fabrication process, especially for the sub-nanometer resolution. In multiple rings design ([14] in the reference), the tuning of resonance wavelength is realized by varying the outer radius with 1 nm step to achieve 0.6 nm resonance wavelength increment. It is difficult to control the dimension tolerance within 1 nm because imperfect fabrication and/or non-uniform film thickness of core layer will result in a large deviation (tens of nanometers) from the actual design.

Comment 5: *Regarding the measurements, the experiments do not include a broadband signal measurement, only two fairly closely spaced narrowband laser lines are used for measurements. This is not sufficient to justify publication in high impact journal.*

Reply: As suggested by the reviewer, a broadband signal measurement is added to revised manuscript with the result shown in Fig. 6c. The description of measurement process and result is added to revised manuscript as “To further evaluate the performance of the RAFT spectrometer, we performed a broadband signal measurement. A broadband ASE light source (Amonics ALS-CL) covering C and L band is used as the input. For broadband signal input, all the detected input sparse spectra are retrieved using the normalization coefficient matrix A (see Supplementary Fig. S9) and are then combined to produce the original input spectrum. The retrieved spectrum and input broadband source are shown in the same figure (Fig. 6c) for comparison. The retrieved spectrum agrees with the input spectrum very well. The small discrepancy is due to misalignment between lensed fiber and inverse-taper waveguide coupler and resonance position variation due to thermal crosstalk. By packaging the lensed fiber to the input waveguide, the misalignment would not be present. The thermal crosstalk mainly originates from silicon substrate since the buried oxide layer (BOX) is not thick

enough to effectively isolate the heat from the heater above the MZI to Si substrate (as in our experiment, the BOX is 2 μ m). The residue thermal crosstalk can be mitigated through fabrication and optimization of isolation trenches near MZI arms. It will also be reduced by making both MRR and MZI fully suspend and/or using thicker BOX. Moreover, it will be well compensated with a feedback control circuit.” in Line 253 Page 13.

Fig. 6c: Normalized retrieved spectrum (black) with ASE broadband source input (red).

In fact, this is not a surprise, as the authors neglect dispersion in their analysis, which would spoil an attempt to characterize a broadband input signal. The authors also neglect the benefit of dispersion in estimating the limits of resolution/temperature efficiency for a pure FTIR spectrometer. This is somewhat disappointing, making the device performance look better.

Reply: As pointed out by the reviewer, the dispersion is considered for broadband input signal measurement since silicon has a strong dispersion in the working bandwidth. By referring to [26] in the reference, the description of dispersion analysis, calculation and related parameter values are added to revised Supplementary Note 2. Waveguide dispersion contributes to stretched retrieved spectrum around ν_0 .

In a tunable MZI-based FTIR spectrometer, the resolution is determined not only by the maximal optical path difference, but also by the dispersion in the waveguide considering the wide bandwidth. The resolution of the tunable MZI is improved with decreasing wavelength. The resolution limit analysis due to dispersion is added in revised manuscript as “The resolution limit of the tunable MZI due to waveguide dispersion is $18 \leq R \leq 19.9$ nm in the detected wavelength range with fixed maximal OPD (with maximal heating power of 1.8 W employed in the experiment).” in Line 284 Page 14.

Comment 6: *The calibration procedure is not clearly explained. Was the calibration performed independently for each wavelength? How will it be done for operation with broadband signals?*

Reply: As suggested by the reviewer, the calibration process for broadband signal is added to revised Supplementary Note 2 as “A broadband ASE light source (Amonics ALS-CL) covering C and L band is used as the input. Here in our experiment, the bandwidth is 90 nm and FSR of the MRR is ~28 nm. Hence the number of the retrieved resolution elements is 3 for each tuning state of the MRR. After completing the MRR thermal tuning to cover one FSR, all the sampled interferograms from input source are obtained. There are two steps to perform wavelength/frequency calibration. The first step is to coarsely determine the frequencies of the sparse spectra by performing FFT to the sampled interferograms. The second step is to finely determine the frequencies according to the tuning state of the MRR. Subsequently, we normalize the retrieved power to the input power for each wavelength to obtain a normalization coefficient matrix A including MRR tuning states, wavelengths and their corresponding transmission coefficients (Supplementary Fig. S9). Hence, for broadband signal input, all the detected sparse spectra are retrieved using the normalization coefficient matrix A and then the retrieved spectra are combined to produce the original input spectrum.” in Line 161 Page 11.

Supplementary Fig. S9: Normalization coefficient matrix A including MRR tuning states, wavelengths and their corresponding transmission coefficients.

Comment 7: *There are a number of spelling errors throughout the manuscript, and the authors should check to make sure that the units are capitalized where need be ($k \rightarrow K$ for Kelvins, etc).*

Reply: As pointed out by the reviewer, the “ k/W ” is corrected in revised manuscript as “ K/W ” in Line 213 Page 11. Other errors are also corrected in revised manuscript.

Reply to Reviewer 2

We are grateful to the Reviewer for the constructive comments and are happy to address all the comments.

Comment 1: *By using the ring resonator as a filter, it is essential that there is no change in the transmission value for the different resonances at a given temperature as well as no variation as a function of the temperature tuning. The characterisation of the resonances and the transmission properties of the input ring resonator have to be addressed both theoretically and experimentally.*

Reply: As pointed out by the reviewer, due to the large material dispersion of silicon, the ring couplers are wavelength dependent. Hence at a given temperature, different resonance wavelengths have different transmission values, which will also change with the temperature tuning. To address this issue, we take the ring and MZI as a whole, and employ a transmission coefficient matrix to calibrate the power for each wavelength. The calibration process for broadband signal is added in revised Supplementary Note 2 as “A broadband ASE light source (Amonics ALS-CL) covering C and L band is used as the input. Here in our experiment, the bandwidth is 90 nm and FSR of the MRR is ~28 nm. Hence the number of the retrieved resolution elements is 3 for each tuning state of the MRR. After completing the MRR thermal tuning to cover one FSR, all the sampled interferograms from input source are obtained. There are two steps to perform wavelength/frequency calibration. The first step is to coarsely determine the frequencies of the sparse spectra by performing FFT to the sampled interferograms. The second step is to finely determine the frequencies according to the tuning state of the MRR. Subsequently, we normalize the retrieved power to the input power for each wavelength to obtain a normalization coefficient matrix A including MRR tuning states, wavelengths and their corresponding transmission coefficients (Supplementary Fig. S9). Hence, for broadband signal input, all the detected sparse spectra are retrieved using the normalization coefficient matrix A and then the retrieved spectra are combined to produce the original input spectrum.” in Line 161 Page 11.

Supplementary Fig. S9: Normalization coefficient matrix A including MRR tuning states, wavelengths and their corresponding transmission coefficients.

Comment 2: *The authors mention the heat isolation trenches to improve the heating/thermal optical effect efficiency. Thermal crosstalk between the MZ and the ring resonator is also a major problem, because the operation of the former might affect the resonance position of the ring resonator. This is not addressed neither theoretically nor experimentally in the manuscript*

Reply: As suggested by the reviewer, the simulation result of the isolation trenches to improve heating efficiency is presented in Supplementary Note 1 and shown in

Supplementary Fig. S2a. MZI testing structures (shown in Supplementary Fig. S11a) are fabricated to experimentally test the influence of isolation trenches on the heating efficiency and the results are added in revised Supplementary Note 3. The heating efficiency can be improved to maximum 12 times by reducing the gap and/or increasing the trench segment length L . The discussion on the heating efficiency improvement is added in revised manuscript as “The total energy consumed by MRR and MZI for N scans are 1.96 J and 67.2 J, respectively. The calculation is presented in Supplementary Note 3. The power consumption of MZI can be reduced to 150 mW through fabrication of isolation trenches along the waveguides of MZI arms (see Supplementary Note 3). Hence, the total energy consumed by MZI can be reduced from 67.2 J to 5.6 J. Note that the Si substrate under the MRR in the tested RAFT spectrometer and the MZI arm in the testing structure is not totally removed (Supplementary Fig. S11b), the heating efficiency of both MRR and MZI can be further reduced (~ 8.75 times) if the waveguides are fully suspended^[31]” in Line 276 Page 14.

Supplementary Fig. S2a: Relation between the static temperature and the heating power with and without trenches.

Supplementary Fig. S11: **a.** Schematic of MZI testing structure with isolation trenches. **b.** SEM image of the cross section of the thermal isolation trench. **c.** Power consumption vs gap between the trench and waveguide.

As pointed by the reviewer, thermal crosstalk will affect the resonance position of the ring. The thermal crosstalk effect and thermal compensation method to stabilize the resonance position are added in revised Supplementary Note 3. The thermal crosstalk from MZI heater 2 will affect the resonance position of the MRR (Supplementary Fig. S12), while there is no obvious influence on the MZI by MRR heater 1. The maximum resonance wavelength shift due to thermal crosstalk from the MZI heater is 1 nm. The thermal compensation equation is presented in Supplementary Eq. 29 as

$$P_r = \frac{\Delta\lambda_n}{B_1} + \frac{B_2}{B_1} (P_{m-\max} - P_m) \quad (2)$$

With thermal compensation, the thermal crosstalk has been well compensated as shown in Supplementary Fig. S13. More interference information is obtained after compensation compared to that before compensation.

Supplementary Fig. S12: Relation between resonance wavelength shift and heating power on heater 2 due to thermal crosstalk.

Supplementary Fig. S13: Interferograms with a tunable laser source input **a** before and **b** after thermal compensation.

Comment 3: *In the thermal modulation session, absolutely no non-linearity in the thermal optical effect (also temperature expansion) is considered. The analysis is very simplistic. In fact, given the excursion of the MZ thermal optical effect (1.8 W and temperatures up to 150 C) it is essential that non-linearities are taken care of as it has been in recent published in Nature Comm on the same subject.*

Reply: As pointed out by the reviewer, nonlinearity exists in the thermo-optic coefficient (TOC) and thermal expansion. By referring to [26] in the reference, we calculated parameters relating to waveguide dispersion, thermo-optic effect and thermal expansion as shown in the Supplementary Table 1. The analysis is added in revised Supplementary Note 2. TOC nonlinearity and thermal expansion broaden and shift the spectrum to higher frequencies. Waveguide dispersion contributes to stretched retrieved spectrum around ν_0 .

Parameter	Value	Unit	Parameter	Value	Unit
$n_{eff} _{v_0}$	2.23	-	$\partial_{T^2} n$	2.5×10^{-7}	K^{-2}
$\partial_v n$	1.1×10^{-2}	THz^{-1}	$\partial_{v,T^2} n$	-4.6×10^{-9}	$\text{K}^{-2} \text{THz}^{-1}$
$\partial_{v^2} n$	4.8×10^{-6}	THz^{-2}	$\partial_{v^2,T^2} n$	1.7×10^{-9}	$\text{K}^{-2} \text{THz}^{-2}$
$\partial_{v^3} n$	-2.3×10^{-6}	THz^{-3}	$\partial_{v^3,T^2} n$	7.0×10^{-10}	$\text{K}^{-2} \text{THz}^{-3}$
Parameter	Value	Unit	Parameter	Value	Unit
$\partial_T n$	1.9×10^{-4}	K^{-1}	α_1	2.5×10^{-6}	K^{-1}
$\partial_{v,T} n$	3.5×10^{-7}	$\text{K}^{-1} \text{THz}^{-1}$	α_2	8.5×10^{-9}	K^{-1}
$\partial_{v^2,T} n$	-6.4×10^{-8}	$\text{K}^{-1} \text{THz}^{-2}$	α_3	-2.3×10^{-11}	K^{-1}
$\partial_{v^3,T} n$	-2.0×10^{-8}	$\text{K}^{-1} \text{THz}^{-3}$			

Supplementary Table 1: Parameter values of waveguide dispersion, thermo-optic effect and thermal expansion. The dispersion and thermo-optic coefficients are obtained for the quasi-TE mode of waveguide shown in Supplementary Fig. S1. The partial derivative is used as $\partial n_{eff} / \partial x = \partial_x n$.

Comment 4: *In the single wavelength characterisation, there are several important issues to be considered.*

4.1) *Firstly, the acronym TLS is never defined. I assume is tunable laser source. This is indeed simple to be corrected*

Reply: As pointed out by the reviewer, *TLS* is indeed the abbreviation of *tunable lase source*. The definition is added in revised manuscript as “A tunable laser source (TLS-1: Santec TSL-510) is used for single wavelength characterization.” in Line 204 Page 10.

4.2) *There is a reference to a resonance shift that is indeed very confusing. What is this shift? Is it caused by the input light power? The entire discussion of the resonance shift which is done onwards is very difficult to follow. This has to be very, very, well clarified.*

Reply: As pointed by the reviewer, the resonance wavelength shift $\Delta\lambda$ is the shift with respect to the initial resonance wavelength λ_0 . The following sentences are added in revised manuscript as “Here for simplicity, we choose the smallest resonance wavelength of the MRR in the detected wavelength range to make the following discussion and assume λ_0 as the initial resonance wavelength. When heater 1 is activated by an external voltage, the resonance position of the MRR will shift to λ_r , inducing a relative wavelength shift as $\Delta\lambda = \lambda_r - \lambda_0$.” in Line 95 Page 5.

4.3) *The authors mention that under a single wavelength input one observes a sinusoidal behaviour of the photodetector current, as shown in Fig. 5a. The behaviour is absolutely not sinusoidal. There is a large amplitude variation, plus, although not shown, there should be chirp which would explain the large difference between the retrieved spectrum with respect to the input. This may not appear to be a big problem when a source with only one or two wavelengths are employed. However, this would make the device unusable for a continuous reconstruction. This is the most serious negative issue of this manuscript. There should be a demonstration of a wide spectrum recovery. Based on the results for single or double wavelength, it appears that the non-linearities that I commented above, if not considered and taken care of, will make the device unusable. Therefore, a better evaluation of the non-linearity and a demonstration of a wide spectrum recovery is paramount for the demonstration of the device functionality*

Reply: Ideally for a single wavelength input, there should be a sinusoidal behavior in the photodetector current. As pointed out by the reviewer, due to the dispersion, TOC and thermal expansion non-linearities, the nonlinearity of heating power P axis and amplitude variation are observed in the detected interferogram (detected power intensity vs heating power on MZI). As pointed out by the reviewer, we have deleted “The intensity changes sinusoidally with the applied power on heater 2.” in revised manuscript.

The review is right that there will be chirp with a broadband source input for a typical FT spectrometer. In our proposed design, input broadband source is filtered first to obtain multiple discrete resonance wavelengths, which are then input to the cascaded tunable MZI to resolve. Therefore, the chirp is not observed in our experiment.

As suggested by the reviewer, the analysis of waveguide dispersion, TOC nonlinearity and thermal expansion is added in revised Supplementary Note 2. As shown in Supplementary Eq. 12-16, TOC nonlinearity and thermal expansion broaden and shift the spectrum to higher frequencies and waveguide dispersion contributes to stretched retrieved spectrum around ν_0 .

As suggested by the reviewer, we have performed the wide spectrum measurement as shown in Fig. 6c. The description of measurement process and result is added in revised manuscript as “To further evaluate the performance of the RAFT spectrometer, we performed a broadband signal measurement. A broadband ASE light source (Amonics ALS-CL) covering C and L band is used as the input. For broadband signal input, all the detected input sparse spectra are retrieved using the normalization coefficient matrix A (see Supplementary Fig. S9) and are then combined to produce the original input spectrum. The retrieved spectrum and input broadband source are shown in the same figure (Fig. 6c) for comparison. The retrieved spectrum agrees with the input spectrum very well. The small discrepancy is due to misalignment between lensed fiber and inverse-taper waveguide coupler and resonance position variation due to thermal crosstalk. By packaging the lensed fiber to the input waveguide, the misalignment would not be present. The thermal crosstalk mainly originates from silicon substrate since the buried oxide layer (BOX) is not thick enough to effectively isolate the heat from the heater above the MZI to Si substrate (as in our experiment, the BOX is 2 μm). The residue thermal crosstalk can be mitigated through fabrication and optimization of isolation trenches near MZI arms. It will also be reduced by making both MRR and MZI

fully suspend and/or using thicker BOX. Moreover, it will be well compensated with a feedback control circuit.” in Line 253 Page 13.

Fig. 6c: Normalized retrieved spectrum (black) with ASE broadband source input (red).

Reply to Reviewer 3

We are grateful to the Reviewer for the constructive comments and are happy to address all the comments.

Comment 1: *The term "Cavity-enhanced" in the title is not used in its conventional manner. The enhancement cavity in spectroscopy is generally used for increasing the light-matter interaction length and thus boosting the sensitivity of the spectrometer, while the authors used this term to state the enhancement in the spectral resolution of the spectrometer. I personally find this misleading and strongly suggest either to remove it from the title or to rephrase it.*

Reply: As suggested by the review, we have rephrased the title in revised manuscript as "A Microring Resonator-assisted Fourier Transform Spectrometer with Enhanced Resolution and Large Bandwidth in Single Chip Solution"

Comment 2: *Although the idea of the authors in cascading a MRR with a MZI to increase the spectral resolution seems to be novel, unfortunately they fail to demonstrate this enhancement in practice. In Fig. 4 (a) the authors show the transmission spectrum of the MRR with ~ 0.5 nm FWHM, measured with an optical spectrum analyzer, not the MZI. In Fig. 6 the measured spectrum of two cw laser sources (with a wavelength difference of ~ 30 nm) are shown and it is obvious from the figure that the spectral resolution of the spectrometer is ~ 20 nm. The authors do not demonstrate the possibility of resolving two spectral features with a separation in the order of their claimed resolution (~ 0.5 nm). In addition the process of extracting a high resolution broadband spectrum by their spectrometer is not explained and there is no comments about the time scale needed for performing such a measurement. The measurement time is specifically important since the performance of a heater-based spectrometer is compared with an AWG-based one (see [5-8]).*

Reply: The working principle of the RAFT spectrometer is that the MRR produces sparse spectra at each tuning state for retrieval by the tunable MZI. Firstly, the requirement for the MRR is that at tuning state $\Delta\lambda_n$, the resonance peaks can suppress the resonance peaks of the adjacent tuning states, i.e. $\Delta\lambda_{n-1}$ and $\Delta\lambda_{n+1}$ (both are off-resonance at current tuning state $\Delta\lambda_n$) as shown in Fig. 5b. This defines the minimum tuning value of the MRR, i.e. the final resolution. Secondly, the tunable MZI must distinguish the 3 resonance peaks at each tuning state as shown in Fig. 6a-b. This verifies that the resolution of the tunable MZI is sufficient to differentiate the sparse spectrum from MRR.

As suggested by the reviewer, we have performed broadband source retrieval shown in Fig. 6c. The description of measurement process and result is added in revised manuscript as "To further evaluate the performance of the RAFT spectrometer, we performed a broadband signal measurement. A broadband ASE light source (Amonics ALS-CL) covering C and L band is used as the input. For broadband signal input, all the detected input sparse spectra are retrieved using the normalization coefficient matrix A (see Supplementary Fig. S9) and are then combined to produce the original input spectrum. The retrieved spectrum and input broadband source are shown in the same figure (Fig. 6c) for comparison. The retrieved

spectrum agrees with the input spectrum very well. The small discrepancy is due to misalignment between lensed fiber and inverse-taper waveguide coupler and resonance position variation due to thermal crosstalk. By packaging the lensed fiber to the input waveguide, the misalignment would not be present. The thermal crosstalk mainly originates from silicon substrate since the buried oxide layer (BOX) is not thick enough to effectively isolate the heat from the heater above the MZI to Si substrate (as in our experiment, the BOX is 2 μ m). The residue thermal crosstalk can be mitigated through fabrication and optimization of isolation trenches near MZI arms. It will also be reduced by making both MRR and MZI fully suspend and/or using thicker BOX. Moreover, it will be well compensated with a feedback control circuit.” in Line 253 Page 13.

Fig. 6c: Normalized retrieved spectrum (black) with ASE broadband source input (red).

As suggested by the reviewer, we have performed the time scale measurement and the results are added in revised Supplementary Note 4 as “The time-scale measurement is performed. For MZI thermal response, the rise and fall time are 37 μ s and 60 μ s, respectively, as shown in Supplementary Fig. S15a. For MRR thermal response, the rise and fall time are 20 μ s and 80 μ s, respectively, as shown in Supplementary Fig. S15b.” in Line 285 Page 19.

The discussion on the time is added in revised manuscript as “Nearly 2% ($\delta\lambda/FSR$) of total time is spent for measuring a single resolution element and in our current experiment, the one-time scan duration is 2 s. The time-scale measurement on thermal response time of MZI and MRR is presented in Supplementary Note 4. The results show that the maximum sweeping frequency of MZI is 10 kHz. For 10 kHz sampling frequency and 2,000 one-time sampling points, the one-time scan duration is reduced to 0.2 s and the total time is reduced to $0.2FSR/\delta\lambda \approx 11.4$ s.” in Line 310 Page 15.

Supplementary Fig. S15: Thermal response time of **a** tunable MZI and **b** tunable MRR.

Comment 3: *There are several grammatical and phrasing errors in the text that at some cases even make it hard to understand the general point. I strongly recommend the authors to ask a native speaker to review and refine the manuscript.*

Reply: As pointed out by the reviewer, we have corrected the errors in revised manuscript.

Reviewers' Comments:

Reviewer #1:

Remarks to the Author:

The authors did a significant revision and responded to all of the comments I raised in the review. One last concern is related to operating bandwidth and issues that some of the components may have. Specifically:

1. The main problem with the evanescent couplers is their bandwidth of operation. Additionally, they have to be very finely controlled to get the small coupling coefficients that are needed to make high Q resonators. How Q will depend on the coupling coefficient and in turn what fabrication tolerance will need to be achieved?
2. There also is an issue of stabilization that is characteristic to ring resonators. What needs to be the stability tolerance and how it can be achieved?
3. Moreover, the SNR is inherently an issue with the described approach and should be clearly stated in abstract and conclusions

Reviewer #2:

Remarks to the Author:

I believe the authors have properly addressed all of my comments, and I believe the manuscript is adequate for publishing. As a minor suggestion, I would like to make two requests:

- (1) In the abstract..remove the .."For the first time ...". If the work is novel, it is the first time!
- (2) Figure S2a has a typo: "piont"s instead of "points".

Newton C. Frateschi

Reviewer #3:

Remarks to the Author:

The authors have updated the manuscript by adding more details about the measurement process and the developed instrument, rephrasing the title, presenting a more detailed discussion about the performance of the instrument and a lengthy supplementary note. All of these modifications helped to elevate the quality of the article.

The authors also managed to measure broadband spectrum of an ASE source in Fig. 6 (c) with, apparently, a spectral point spacing of ~ 0.5 nm. This is definitely a forward step in showing the capability of their instrument to measure a broadband spectrum; however, they again fail to demonstrate the possibility of resolving two spectral features with a separation in the order of their claimed resolution (~ 0.5 nm). The structures on the ASE spectral profile in Fig. 6 (c) (black curve) could be due to the artifacts that the authors mention in the revised manuscript and since the measured ASE spectrum in Fig. 6 (c) (red curve, which they compare their measurement to) is quite low resolution, it would not help to evaluate the spectral resolution of their instrument.

I suggest the authors to re-measure the spectra shown in Fig. 6 (a) and (b) when the frequency of the two tunable laser sources are only separated by 0.5 nm and show the ability to resolve them.

In addition, to show that the instrument can offer this spectral resolution for the entire 90 nm bandwidth (as they claim in the abstract), I suggests the authors to measure a broadband absorption spectrum of a (or multiple) gas species, e.g. CO₂. The fine absorption lines of gas phase species are spread around in a broad spectral range, which would nicely verify the performance of the "spectrometer with enhanced resolution and large bandwidth", as it is claimed in the title of the manuscript.

In conclusion, despite of all improvements, unfortunately I believe that the manuscript – on its present status – is not yet appropriate to be published in nature communications since the main claim of the article is not yet shown to be achieved.

Manuscript ID: NCOMMS-18-23000B

Paper title: A Microring Resonator-assisted Fourier Transform Spectrometer with Enhanced Resolution and Large Bandwidth in Single Chip Solution

Authors: S. N. Zheng, H. Cai, J. F. Song, L. K. Chin, P. Y. Liu, Z. P. Lin, D. L. Kwong, and A. Q. Liu

Reply to Reviewer 1

We are grateful to the Reviewer for the constructive comments and are delighted that the Reviewer recommended for publication of this manuscript. We are happy to address all the comments.

Comment 1: *The main problem with the evanescent couplers is their bandwidth of operation. Additionally, they have to be very finely controlled to get the small coupling coefficients that are needed to make high Q resonators. How Q will depend on the coupling coefficient and in turn what fabrication tolerance will need to be achieved?*

Reply: As pointed out by the reviewer, the bandwidth of the designed ring coupler is ~ 100 nm. Hence, to extend the spectral range, multiple RAFT spectrometer elements need to be implemented. The discussion is added in revised manuscript as “A high-resolution of 0.47 nm and a large bandwidth of ~ 90 nm is achieved. The bandwidth can be largely extended by integrating a paralleled RAFT element array.” in Line 343 Page 16.

As suggested by the reviewer, the analysis and discussion on how Q depends on coupling coefficient are added in revised Supplementary Note 5 as

“The quality factor Q value for a symmetric add-drop ring is expressed as^[9]

$$Q = \frac{\pi n_g L \sqrt{r^2 a}}{\lambda_r (1 - r^2 a)} \quad (33)$$

where n_g is the group index, L is the round-trip length of the MRR, r is the self-coupling coefficient, and a is the single-pass amplitude transmission.

The Q value is proportional to r as shown in Supplementary Fig. S16. The values of each parameter in the simulation are shown in Supplementary Table S3.” in Line 295 Page 20.

Supplementary Fig. S16: Relation between Q value and self-coupling coefficient r .

The paper [9] is added into reference in revised supplementary material in Line 327 Page 23 as

[9] W. Bogaerts, P. De Heyn, T. Van Vaerenbergh, K. De Vos, S. Kumar Selvaraja, T. Claes, P. Dumon, P. Bienstman, D. Van Thourhout, and R. Baets, "Silicon microring resonators," *Laser & Photonics Reviews* 6, 47-73 (2012).

Parameter	n_g	λ_r	L	a
value	4.25	1528.256 nm	20.734 μm	0.9986

Supplementary Table S3: Parameters values for simulating relation between Q value and r .

As suggested by the reviewer, the discussion on the fabrication tolerance is added in revised manuscript as “For instance, for single-pass amplitude transmission $a = 0.9986$, if $Q \geq 10,000$, i.e., $\delta\lambda \leq 0.153$ nm at 1530 nm, the self-coupling coefficient $r \geq 0.9835$ (see Supplementary Note 5). Hence, the gap between ring and straight waveguide is larger than 230 nm. Only the fabrication tolerance of gap is considered here, since the gap dominates in determining r , thereby, the Q value. Considering that the transmitted power will be reduced when increasing Q value, the designed gap is 250 nm with 20 nm fabrication tolerance. The current fabrication tolerance for gap and linewidth is 15 nm, which is enough for 20-nm tolerance.” in Line 299 Page 14.

Comment 2: *There also is an issue of stabilization that is characteristic to ring resonators. What needs to be the stability tolerance and how it can be achieved?*

Reply: As pointed out by the reviewer, the discussion on the stability tolerance is added in revised manuscript as “By employing isolation trenches around MRR and thermal compensation (see Supplementary Note 3), the stability tolerance of the resonance wavelength has been decreased from $2\delta\lambda$ (without thermal compensation) to $\delta\lambda/5$. The current value can be further decreased by reducing residue thermal crosstalk, optimized thermal compensation and/or adopting heater with low temperature coefficient of resistance. The residue thermal crosstalk can be further mitigated through fabricating isolation trenches near MZI arms and can also be effectively reduced by making both MRR and MZI fully suspended^[32] (see Supplementary Note 3) and/or using thicker BOX. Moreover, it will be well compensated with a feedback circuit to control the applied power on MRR and MZI heater.” in Line 269 Page 13.

Comment 3: *Moreover, the SNR is inherently an issue with the described approach and should be clearly stated in abstract and conclusions*

Reply: As suggested by the reviewer, the statements on SNR are added in revised manuscript as

“The MRR boosts the resolution to 0.47 nm, which is far beyond the Rayleigh criterion of the tunable MZI-based Fourier-transform (FT) spectrometer. A single channel integrated with a single PD can achieve a large bandwidth of ~ 90 nm with low power consumption (35 mW for MRR heater and 1.8 W for MZI heater) at the expense of degraded SNR due to time-multiplexing.” in Line 19 Page 1.

“The SNR degraded by time-multiplexing can be improved by reducing optical loss and/or adopting smaller FSR.” in Line 347 Page 17.

In summary, we have addressed all the comments from Reviewer 1. The manuscript and supplementary material have been carefully corrected.

Reply to Reviewer 2

We are grateful to the Reviewer for the constructive comments and are delighted that the Reviewer recommended the publication of this manuscript. We are happy to address all the comments.

Comment 1: *In the abstract..remove the .."For the first time ...". If the work is novel, it is the first time!*

Reply: As pointed out by the reviewer, "for the first time" is removed from the abstract and conclusion in revised manuscript in Page 1 and Page 16, respectively.

Comment 2: *Figure S2a has a typo: "piont"s instead of "points".*

Reply: As pointed out by the reviewer, the typo "pionts" is corrected as "points" in revised Supplementary Fig.S2a in Page 3.

Supplementary Fig. S2a: Relation between the static temperature and the heating power with and without trenches.

In summary, we have addressed all the comments from Reviewer 2. The manuscript and supplementary material have been carefully corrected.

Reply to Reviewer 3

We are grateful to the Reviewer for the constructive comments and are happy to address all the comments.

Comment 1: *The authors also managed to measure broadband spectrum of an ASE source in Fig. 6 (c) with, apparently, a spectral point spacing of ~ 0.5 nm. This is definitely a forward step in showing the capability of their instrument to measure a broadband spectrum; however, they again fail to demonstrate the possibility of resolving two spectral features with a separation in the order of their claimed resolution (~ 0.5 nm). The structures on the ASE spectral profile in Fig. 6 (c) (black curve) could be due to the artifacts that the authors mention in the revised manuscript and since the measured ASE spectrum in Fig. 6 (c) (red curve, which they compare their measurement to) is quite low resolution, it would not help to evaluate the spectral resolution of their instrument. I suggest the authors to re-measure the spectra shown in Fig. 6 (a) and (b) when the frequency of the two tunable laser sources are only separated by 0.5 nm and show the ability to resolve them.*

Reply: As pointed out by the reviewer, the working process of the microring resonator-assisted Fourier-transform (RAFT) spectrometer is that the tunable MZI distinguishes the sparse spectrum (3 resonance peaks) from MRR at each MRR tuning state. The intention is to verify that the resolution of the tunable MZI is sufficient to differentiate the sparse spectrum from MRR. The original spectrum is obtained by combining the reconstructed sparse spectra at all tuning states. The description and measurement results are modified in revised manuscript as “TLS-1 and TLS-2 (ANDO AQ4321D) are combined with a 50/50 optical coupler as the input for double wavelength characterization. Figure 6a shows the retrieved spectra when TLS-1 and TLS-2 are set at adjacent on-resonance wavelengths. The tuning state is resonance wavelength shift $\Delta\lambda = 0$. The resonance wavelengths are $\lambda_{on1} = 1525.400$ nm, $\lambda_{on2} = 1552.844$ nm, and $\lambda_{on3} = 1581.240$ nm, respectively. One can see that the two adjacent on-resonance wavelength components can be easily distinguished and reconstructed by the tunable MZI. As a result, the tunable MZI can retrieve each filtered spectrum at each tuning state of the MRR.” in Line 242 Page 12.

Fig. 6a: Retrieved spectra with TLS-1 and TLS-2 (set at adjacent on-resonance wavelengths of MRR, respectively) input simultaneously when $\Delta\lambda = 0$.

As suggested by the reviewer, we performed many experiments to verify the resolution. It should be noted that by referring to [6] and [14], where the resolutions are typically defined as the channel spacing or linewidth of ring resonance peak, respectively, we define our

resolution based on how well the device will filter out the off-resonance wavelength components. We firstly test the resolution of the RAFT spectrometer using a tunable laser source (TLS-1: Santec TSL-510). The descriptions on measurement and results are added in revised manuscript as “We test the resolution of the RAFT spectrometer using TLS-1. Here, we employ three resonance peaks ($\lambda_{on1} < \lambda_{on2} < \lambda_{on3}$) of the MRR to filter the input source. For simplicity, we define a detuning wavelength $d\lambda$ as $\lambda_{off} - \lambda_{on}$ indicating the difference between off-resonance wavelength λ_{off} and on-resonance wavelength λ_{on} . We compare the retrieved power intensity of λ_{on} and λ_{off} after FFT. The power of TLS-1 is set at 8 mW. The MRR is tuned to $\Delta\lambda = 3.38$ nm. Figure 5b shows the retrieved spectra when $\lambda_{on3} = 1584.620$ nm and the value of $d\lambda$ is set to be 0.3 nm and 0.47 nm, respectively. One can see that when $d\lambda = 0.3$ nm, the retrieved power ratio between the on-resonant wavelength λ_{on} and the detuned off-resonant wavelength λ_{off} equals to 6.85 dB, while when $d\lambda = 0.47$ nm, the retrieved power ratio is increased to 10.10 dB. Similarly, the retrieved power ratios at $d\lambda = 0.47$ nm for the other two resonance wavelengths (i.e., $\lambda_{on1} = 1528.488$ nm and $\lambda_{on2} = 1556.020$ nm) are 14.77 dB and 16.46 dB, respectively. To effectively filter out the detuned off-resonant components into the drop port, we define the retrieved power ratio should be larger than 10 dB, i.e., the minimum MRR tuning value is 0.47 nm. Hence, the resolution of the RAFT spectrometer is defined as 0.47 nm.” in Line 222 Page 11.

Fig. 5b: Retrieved spectra with TLS-1 input at 8 mW input power when $\Delta\lambda = 3.38$ nm. The on-resonance wavelength $\lambda_{on3} = 1584.620$ nm and the value of $d\lambda$ is set to be 0.3 nm and 0.47 nm, respectively. (On-resonance wavelengths in black and off-resonance wavelengths in red)

The two reference papers [6] and [14] are added into the references in revised manuscript in Line 414 Page 20 and Line 428 Page 21, respectively as

- [6] P. Cheben, J. H. Schmid, A. Del age, A. Densmore, S. Janz, B. Lamontagne, J. Lapointe, E. Post, P. Waldron, and D. X. Xu, "A high-resolution silicon-on-insulator arrayed waveguide grating microspectrometer with sub-micrometer aperture waveguides," *Opt Express* 15, 2299-2306 (2007).
- [14] Z. Xia, A. A. Eftekhar, M. Soltani, B. Momeni, Q. Li, M. Chamanzar, S. Yegnanarayanan, and A. Adibi, "High resolution on-chip spectroscopy based on miniaturized microdonut resonators," *Opt Express* 19, 12356-12364 (2011).

As suggested by the reviewer, we further test the resolution using a wavelength-division multiplexer (Sharetop WDM) to generate a spectrum with two broad spectral peaks by referring to [25] and [31]. The descriptions of measurement and results are added in revised manuscript as “To further test the resolution, we use a wavelength-division multiplexer (Sharetop WDM) to generate two broad spectral peaks as the input spectrum of the RAFT spectrometer^[25, 31]. The normalized retrieved spectra and the input spectra are shown in Fig. 6b. It can be seen that the minimum resolvable wavelength detuning is 0.448 nm, which is smaller than the minimum MRR tuning value of 0.47 nm. Hence, the resolution is 0.47 nm, which significantly outperforms the Rayleigh criterion of the tunable MZI (19.32 nm).” in Line 251 Page 12.

Fig. 6b: Normalized retrieved spectra (black) using the spectrum with two spectral peaks input (red).

The two reference papers [25] and [31] are added into the references in revised manuscript in Line 449 Page 22 and Line 458 Page 22, respectively as

[25] P. J. Bock, P. Cheben, A. V. Velasco, J. H. Schmid, A. Delâge, M. Florjańczyk, J. Lapointe, D. X. Xu, M. Vachon, and S. Janz, "Subwavelength grating Fourier - transform interferometer array in silicon - on - insulator," *Laser & Photonics Reviews* 7, L67-L70 (2013).

[31] D. M. Kita, B. Miranda, D. Favela, D. Bono, J. Michon, H. Lin, T. Gu, and J. Hu, "High-performance and scalable on-chip digital Fourier transform spectroscopy," *Nature Communications* 9, 4405 (2018).

Comment 2: *In addition, to show that the instrument can offer this spectral resolution for the entire 90 nm bandwidth (as they claim in the abstract), I suggests the authors to measure a broadband absorption spectrum of a (or multiple) gas species, e.g. CO₂. The fine absorption lines of gas phase species are spread around in a broad spectral range, which would nicely verify the performance of the “spectrometer with enhanced resolution and large bandwidth”, as it is claimed in the title of the manuscript.*

Reply: As suggested by the reviewer, we measured a broadband absorption spectrum of CO₂. Unfortunately, we found that there are no such sharp absorption peaks by CO₂ or other gas

species in the working range of the spectrometer (1530-1620 nm), because the absorption peaks in near infrared range are normally broad and weak due to overtones of N-H, O-H and C-H bonds. By referring to [27] and [31], we use an optical fiber interferometer to produce a broadband spectrum with multiple peaks (2-6 nm wavelength separation) across the entire bandwidth of the RAFT spectrometer as the input broadband spectrum. We replaced the previous experimental results with the improved results as shown in Fig. 6c. The descriptions on the measurement and results are added in revised manuscript as “Furthermore, we performed a broadband signal measurement with the minimum MRR tuning value of 0.47 nm. The transmission spectrum from an optical fiber interferometer is used as the input. For broadband signal input, all the detected input sparse spectra are retrieved using the normalization coefficient matrix A (see Supplementary Fig. S9) and are then combined to produce the original input spectrum. The normalized retrieved spectrum and input broadband source are shown in Fig. 6c. The retrieved spectrum agrees well with the input spectrum. The small discrepancy is due to misalignment between lensed fiber and inverse-taper waveguide coupler while heating MZI. Another reason is resonance position fluctuation due to thermal crosstalk (see Supplementary Note 3).” in Line 257 Page 13.

Fig. 6c: Normalized retrieved spectrum (black) with a broadband source (red) input. The source spectrum is generated from an optical fiber interferometer.

The three reference papers [27] and [31] are added into the references in revised manuscript in Line 453 Page 22 and Line 458 Page 22, respectively as

[27] M. C. Souza, A. Grieco, N. C. Frateschi, and Y. Fainman, "Fourier transform spectrometer on silicon with thermo-optic non-linearity and dispersion correction," *Nature communications* 9, 665 (2018).

[31] D. M. Kita, B. Miranda, D. Favela, D. Bono, J. Michon, H. Lin, T. Gu, and J. Hu, "High-performance and scalable on-chip digital Fourier transform spectroscopy," *Nature Communications* 9, 4405 (2018).

In summary, we have addressed all the comments from Reviewer 3. The manuscript and supplementary materials have been carefully corrected.

Reviewers' Comments:

Reviewer #1:

Remarks to the Author:

1. OK

2. The response is confusing. Does 20 nm tolerance supports linewidth of 15 nm? Is this resolution? How does it affect the 0.47 nm number in the abstract?

3. Just stating that "SNR degraded by time-multiplexing" and that it " can be improved by reducing optical loss and/or adopting smaller FSR" is not answering the concern. I would expect some quantitative relation between the resolution and SNR requirement, especially when dealing with broad bandwidth signals. After all, you want to extract enough power through the filter but at the same time the total broad band signal power has to be transmitted through waveguides and not cause TPA, etc.

Reviewer #3:

Remarks to the Author:

The authors demonstrated the possibility of resolving two spectral features with ~ 0.5 nm separation in new figure 6(b) for a narrow spectral range. They also updated figure 6(c) to a spectrum with finer spectral features, showing the capability of the developed instrument in measuring a broadband spectrum with rather fine structures. In addition they have added more explanation and references to clarify their claim, and also updated the abstract of the paper.

There is a minor issue about their claim in the response letter as they state:

"As suggested by the reviewer, we measured a broadband absorption spectrum of CO₂. Unfortunately, we found that there are no such sharp absorption peaks by CO₂ or other gas species in the working range of the spectrometer (1530-1620 nm), because the absorption peaks in near infrared range are normally broad and weak due to overtones of N-H, O-H and C-H bonds."

This statement is simply incorrect. Acetylene (C₂H₂) has very sharp absorption lines in the working range of their spectrometer with very large absorption line strength. Few centimeters of interaction length with a 100% acetylene gas sample would be enough to achieve a spectrum with high signal to noise ratio. However, since the authors managed to demonstrate their claims in the added figure and the updated one, I am satisfied with the response of the authors.

In conclusion, I believe that the manuscript is now appropriate to be published in nature communications.

Manuscript ID: NCOMMS-18-23000C

Paper title: A Microring Resonator-assisted Fourier Transform Spectrometer with Enhanced Resolution and Large Bandwidth in Single Chip Solution

Authors: S. N. Zheng, J. Zou, H. Cai, J. F. Song, L. K. Chin, P. Y. Liu, Z. P. Lin, D. L. Kwong, and A. Q. Liu

Reply to Reviewer 1

We are grateful to the Reviewer for the constructive comments and are delighted that the Reviewer recommended for publication of this manuscript. We are happy to address all the comments.

Comment 1: *The response is confusing. Does 20 nm tolerance supports linewidth of 15 nm? Is this resolution? How does it affect the 0.47 nm number in the abstract?*

Reply: As pointed out by the reviewer, the 15 nm tolerance refers to the deviation range (± 7.5 nm) of the width of fabricated waveguide from its design value. The tolerance will affect the value of the gap between ring and straight waveguide, which will affect the self-coupling coefficient r , i.e., the Q value and will thereby affect the resolution of the spectrometer.

The discussion on the fabrication tolerance is modified in revised manuscript as “Noticing that the gap dominates in determining r , thereby, the Q value, only the fabrication tolerance of gap is considered here. Since the transmitted power from MRR will be reduced when increasing Q value (see Supplementary Fig. S17a), the designed gap is 240 nm with 20 nm tolerance, i.e., ± 10 nm fabrication deviation, which can be easily achieved by the current fabrication technology (± 7.5 nm deviation).” in Line 304 Page 15.

Comment 2: *Just stating that “SNR degraded by time-multiplexing” and that it “ can be improved by reducing optical loss and/or adopting smaller FSR” is not answering the concern. I would expect some quantitative relation between the resolution and SNR requirement, especially when dealing with broad bandwidth signals. After all, you want to extract enough power through the filter but at the same time the total broad band signal power has to be transmitted through waveguides and not cause TPA, etc.*

Reply: As suggested by the reviewer, quantitative analysis on SNR and resolution is added in revised Supplementary Note 5 as

“The transmission to the drop port is expressed as^[9]

$$T = \frac{(1 - r^2)^2 a}{1 - 2r^2 a \cos \phi + r^4 a^2} \quad (34)$$

where $\phi = 2\pi n_{\text{eff}} L / \lambda$ is the single-pass phase shift. The simulated relation between the transmitted power carried by the filtered sparse spectrum (number of FSR $m = 3$) and resolution with a uniform broadband source input is shown in Supplementary Fig. S17a. The simulated relation between signal-to-noise ratio (SNR) and resolution at different input power values is shown in Supplementary Fig. S17b. In the experiment, the total loss of the input light is 13.92 dB, including on-chip loss of 4.92 dB and off-chip loss of 9 dB. This loss is considered in the simulation shown in Supplementary Fig. S17b. The values of all parameters (n_g , L , and a) for simulations shown in Supplementary Fig. S17 are summarized in Supplementary Table S3. We see that the minimum resolution value is limited by SNR requirement. In our experiment, the signal can be differentiated from noise and successfully

retrieved when $\text{SNR} \geq 3$ dB, which will define the minimum resolution value for each input power values as shown in Supplementary Fig. S17b.” in Line 312 Page 21.

Supplementary Figure S17: **a** Simulated relation between transmitted power carried by the filtered sparse spectrum ($m = 3$) and resolution. **b** Simulated relation between SNR and resolution at different input power values.

Parameter	n_g	λ_r	L	a
value	4.25	1528.256 nm	20.734 μm	0.9986

Supplementary Table S3: Parameters values for simulations in Supplementary Fig. S16 and Supplementary Fig. S17.

In summary, we have addressed all the comments from Reviewer 1. The manuscript and supplementary material have been carefully corrected.

Reviewers' Comments:

Reviewer #3:

None